# Preference Adaptive and Sequential Text-to-Image Generation

**Ofir Nabati** [1]  **Guy Tennenholtz** [1]  **ChihWei Hsu** [1]  **Moonkyung Ryu** [1]  **Deepak Ramachandran** [2]  **Yinlam Chow** [2]  **Xiang Li** [1]  **Craig Boutilier** [1]

## Abstract

We address the problem of interactive text-to-image (T2I) generation, designing a reinforcement learning (RL) agent which iteratively improves a set of generated images for a user through a sequence of prompt expansions. Using human raters, we create a novel dataset of sequential preferences, which we leverage, together with large-scale open-source (non-sequential) datasets. We construct user-preference and user-choice models using an EM strategy and identify varying *user preference types*. We then leverage a large multimodal language model (LMM) and a value-based RL approach to suggest an adaptive and diverse slate of prompt expansions to the user. Our **P**reference **A**daptive and **S**equential **T**ext-to-image **A**gent (PASTA) extends T2I models with adaptive multi-turn capabilities, fostering collaborative co-creation and addressing uncertainty or underspecification in a user's intent. We evaluate PASTA using human raters, showing significant improvement compared to baseline methods. We also open-source our sequential rater dataset and simulated user-rater interactions to support future research in user-centric multi-turn T2I systems.

## 1. Introduction

Advances in text-to-image (T2I) generation, fueled by powerful diffusion models (Croitoru et al., 2023; Gu et al., 2023; Rombach et al., 2022; Saharia et al., 2022; Yang et al., 2023; Yu et al., 2022; Zhang et al., 2023; Liang et al., 2024), have unlocked unprecedented possibilities for image generation, as users can readily translate textual descriptions into stunning visuals. That said, capturing precise user intent remains a major challenge (Wu et al., 2023; Liang et al., 2024). As

such, single-turn T2I generation may fail to encapsulate a user's nuanced and evolving image conception. This is especially true for complex or abstract concepts, where a user's initial prompt may not suffice in achieving a desired visual representation.

Addressing this challenge requires moving beyond the paradigm of one-shot T2I generation towards a more iterative and interactive process (von Rütte et al., 2023; Wang, 2024; Liu et al., 2024), e.g., a collaborative setting where an assistive agent interacts with a user, guiding them through a series of refinements of their initial prompt. This interactive setup could allow for a more nuanced elicitation of the user's intent, gradually shaping the generated images towards a desired outcome.

To this end, we introduce a **P**reference **A**daptive and **S**equential **T**ext-to-image **A**gent (PASTA), which learns from user preference feedback to guide the user through a sequence of prompt expansions, iteratively refining the generated image. This sequential approach (see Figure 1) allows users to articulate their vision more completely by gradually reducing uncertainty or underspecification in their original prompt. Our framework leverages the power of large multimodal language models (LMMs) (Gemini-Team, 2024; Achiam et al., 2023) and reinforcement learning (RL) (Sutton, 2018; Fan et al., 2023) to facilitate user-adaptive co-creation.

Our approach for PASTA involves a multi-stage data collection and training process. We first collect multi-turn interaction data from human raters with a baseline LMM. Using this sequential data, as well as large-scale, open source (single-turn) preference data, we train a user simulator. Particularly, we employ an EM-strategy to train user preference and choice models, which capture implicit *user preference types* in the data. We then construct a new large-scale dataset, which consists of interactions between a simulated user and the LMM.[1] Finally, we leverage this augmented data, encompassing both human and simulated interactions, to train PASTA – our value-based RL agent, which presents a sequence of diverse slates of images to a user. PASTA

---

[1]Google Research  [2]Google DeepMind.  Correspondence to: Ofir Nabati <ofirnabati@gmail.com>, Guy Tennenholtz <guytenn@gmail.com>.

*Proceedings of the $42^{nd}$ International Conference on Machine Learning*, Vancouver, Canada. PMLR 267, 2025. Copyright 2025 by the author(s).

---

[1]Both human and simulation data is open-sourced to support research on multi-turn T2I generation. Link to the dataset: https://www.kaggle.com/datasets/googleai/pasta-data.

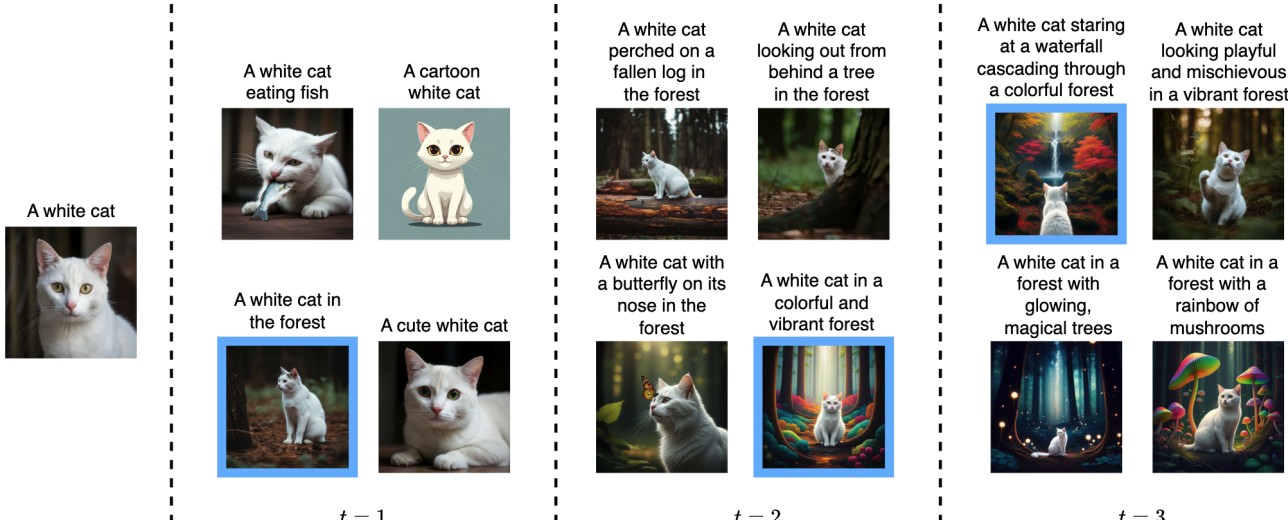

Figure 1: An illustration of an agent-user interaction with $L = 4$ prompt expansions at each step, and $M = 1$ images per prompt expansion. The user selection is outlined in blue. The agent presents prompt expansions based on the user's previous responses to maximizes the expected cumulative user satisfaction (i.e., value). See Appendix G for additional examples using $M = 4$ images for $L = 4$ prompt expansions.

interacts with the user and sequentially refines its generated images to better suit their underlying preferences.

Our contributions are as follows. (1) We formulate the problem of iterative prompt expansion using LMMs and T2I models. (2) We collect a new, large-scale user-agent interaction dataset for the multi-turn prompt expansion setup. Our dataset consists of almost 40,000 interaction rollouts. (3) We train user utility and user-choice models with this dataset, creating a simulator that is used to generate additional simulated data. (4) Finally, we utilize our datasets to train PASTA, our value-based RL agent, demonstrating its effectiveness through comprehensive human evaluations, with significant improvements in user satisfaction compared to a baseline LMM.

## 2. Background and Problem Setup

We begin with a short background on diffusion models, LMMs, and RL. We then formulate our problem setup.

### 2.1. Background

**Diffusion models** (Ho et al., 2020; Ho & Salimans, 2022; Croitoru et al., 2023; Gu et al., 2023; Rombach et al., 2022; Saharia et al., 2022; Yang et al., 2023; Yu et al., 2022; Zhang et al., 2023; Liang et al., 2024) have become a prominent approach for T2I generation. These models add Gaussian noise to an image until it becomes pure noise, and are then trained to reverse this process, iteratively removing noise to reconstruct the image. This denoising process is guided by a prompt, enabling the generation of images that semantically align with the textual description.

**Large Multimodal Models (LMMs)** (Gemini-Team, 2024; Achiam et al., 2023) use Transformer models (Vaswani, 2017), and are trained to predict the next token in a sequence, where tokens can represent textual or visual information. These models learn joint representations of text and images, making them useful for prompt engineering in interactive T2I systems, as they allow us to modify generated outputs based on previously generated responses.

**Reinforcement Learning (RL)** (Sutton, 2018; Fan et al., 2023) frameworks train agents to make a sequence of decisions in an environment to maximize cumulative reward. Value-based RL methods learn a value function that estimates the expected cumulative reward for taking a particular action in a given state. In our work, an RL agent learns to select actions (prompt expansions) that lead to higher user satisfaction.

### 2.2. Problem Formulation

We consider an interactive, sequential T2I decision problem in which a user engages with an agent, visualized in Figure 1. At time $t = 0$, the user issues an initial prompt (e.g., "A white cat") intended to capture a target image. At each turn $t \geq 1$ the agent proposes $L$ prompt expansions, which are fed to a T2I model. The T2I model then generates $M$ images for each prompt. The user sees the slate of $M \times L$ images and selects the set of $M$ images (corresponding to one of the prompt expansions) that best reflects their preferences. The process repeats for up to $H$ turns.

Formally, the problem is given by a tuple $(\mathcal{U}, \mathcal{P}, \mathcal{I}, G, C, R, H, \nu_0)$, where $\mathcal{U}$ is a set of *user types*, reflecting different preferences; $\mathcal{P}$ is a set of feasible

text prompts; $\mathcal{I}$ is a set of feasible images; $G : \mathcal{P} \mapsto \Delta_{\mathcal{I}}$ is a T2I generative model[2]; $C : \mathcal{U} \times \mathcal{P} \times \mathcal{I}^{ML} \mapsto \Delta_{[L]}$ is a *user choice* function[3]; $R : \mathcal{U} \times \mathcal{P} \times \mathcal{I}^{ML} \mapsto \Delta_{[0,1]}$ is a *user utility* function; $H$ is the horizon; and $\nu_0 \mapsto \Delta_{\mathcal{U} \times \mathcal{P}}$ is a distribution of users and initial prompts. At time $t = 0$, a user-prompt pair $u, p_0 \in \mathcal{U} \times \mathcal{P}$ is sampled from the initial distribution $\nu_0$. At each time $1 \le t \le H$ the agent selects a slate of $L$ prompt expansions $P_t = \{p_{\ell,t} \in \mathcal{P}\}_{\ell=1}^{L}$ and feeds them to the T2I model $G$, which generates $M$ images for each prompt, $\left\{ \{I_{m,\ell,t} \sim G(p_{\ell,t})\}_{m=1}^{M} \right\}_{\ell=1}^{L}$. The user observes the slate of $M \times L$ images, and selects a set of $M$ images corresponding to one prompt, and according to the choice function $C$, i.e., $c_t \sim C(u, p_0, I_{1,1,t}, \ldots, I_{M,L,t})$. We assume a (latent) reward $r_t \sim R(u, p_0, I_{1,\ell_t,t}, \ldots, I_{M,\ell_t,t})$ reflecting user $u$'s satisfaction with the selected images.

## 2.3. RL Formulation

Our problem can be formulated as a *latent contextual MDP* (Hallak et al., 2015; Kwon et al., 2021), where $\mathcal{U}$ is the latent context space. The state space consists of the user-agent interaction history, with the state (history) $h_t$ at time $t$ given by:

$$h_t = \left\{ \underbrace{p_0}_{\text{initial prompt}}, \overbrace{\underbrace{\{p_{\ell,1}\}_{\ell=1}^{L}}_{\text{agent action } A_1}, \underbrace{\{I_{m,\ell,1}\}_{\ell=1,m=1}^{L,M}, c_1,}_{\text{transition}}}^{\text{time step 1}} \right. $$
$$\vdots$$
$$\left. \overbrace{\underbrace{\{p_{i,t}\}_{\ell=1}^{L}}_{\text{agent action } A_t}, \underbrace{\{I_{m,\ell,t}\}_{\ell=1,m=1}^{L,M}, c_t}_{\text{transition}}}^{\text{time step } t} \right\},$$

the action space is any selection of $L$ prompts, transitions are induced by the *user choice* model $C$ and the T2I model $G$, and reward is given by a *user utility function* $R$. See Appendix A for a full description of this latent MDP.

A stochastic *policy* $\pi : \mathcal{H} \mapsto \Delta_{\mathcal{P}^L}$ maps interaction histories to a distribution over slates of $L$ prompts. The *value* of a policy $\pi$ is its expected cumulative sum of rewards over users and initial prompts, i.e., $v^{\pi} = \mathbb{E}_{u \sim \nu_0}\left[ \sum_{t=1}^{H} r_t \mid u, \pi \right]$. An *optimal policy* $\pi^* \in \arg\max_{\pi} v^{\pi}$ maximizes the value. The *state-action value function* for any $h \in \mathcal{H}$ and $P \in \mathcal{P}^L$ is $q_t^{\pi}(h, P) = \mathbb{E}_{u \sim \nu_0}\left[ \sum_{t'=t}^{H} r_{t'} \mid u, h_t = h, P_t = P, \pi \right]$.

---

[2]More generally, $G$ can be a function of user $u$ or interaction history. Here we assume $G$ is a *non-adaptive* T2I model.

[3]More generally a user choice function can also depend on prompt expansions. In our work we assume users only view the generated images and the initial prompt.

# 3. PASTA: Preference Adaptive and Sequential Text-to-image Agent

We solve the sequential prompt expansion problem using RL. Our **P**ersonalized **A**nd **S**equential **T**ext-to-image **A**gent (PASTA) engages with a user to adapt the prompt to their preferences, and maximize (latent) user utility. The user's type $u \in \mathcal{U}$ is unknown to the agent and must be inferred during the interaction. This framework is related to meta-RL (Wang et al., 2016; Duan et al., 2016; Finn et al., 2017; Zintgraf et al., 2020), where each episode (or meta-episode) samples a latent MDP from a predefined problem distribution. In our setting, the agent must adapt within $H$ steps to the unknown MDP, and optimize the reward accordingly. This requires balancing exploration and exploitation: the agent must take actions that, on the one hand, provide images that reflect (its estimate of) the user's preferences, and on the other, improving its estimate of those preferences by exploring other types of expansions/images. This can be viewed as an implicit form of *preference elicitation* (Keeney, 1993; Salo & Hamalainen, 2001; Chajewska et al., 2000; Boutilier, 2002; Meshi et al., 2023).

## 3.1. Candidate Action Generator And Selector

The state space of user interaction histories serves as sufficient statistic (i.e., belief over users' types (Aberdeen et al., 2007)). To solve our problem effectively, each interaction with a user should provide the agent sufficient information about the user (e.g., through value of information (Boutilier et al., 2003)). This requires the action space be rich and diverse enough to enable information gain.

A straightforward approach to *action space design* uses an LMM to construct action candidates (in our case, prompt expansion candidates). Specifically, we use an LMM to process the current interaction history and generate a broad set of $L_C \gg L$ candidate prompts from which a slate of $L$ prompts can be chosen. Generating a large set of candidates has been shown to introduce diversity and induce exploration (Tennenholtz et al., 2024). Instead of relying on a given LMM to directly output $L$ prompts, we encourage diversity by generating $L_C$ prompts, and then selecting the $L$-subset our agent deems optimal. This, in turn, expands the effective action space our agent can leverage during training.

Formally, we structure our policy class using a *candidate generator policy* $\pi_C : \mathcal{H} \mapsto \Delta_{\mathcal{P}^{L_C}}$ and a *candidate selector policy* $\pi_S : \mathcal{P}^{L_C} \mapsto \Delta_{\mathcal{P}^L}$. We first sample a set of $L_C$ candidates $\{p_{i,t}\}_{i=1}^{L_C} \sim \pi_C(h_t)$ and then select $L$ prompts from those candidates, i.e., $P_t = \{p_{i,t}\}_{i=1}^{L} \sim \pi_S\left(\{p_{i,t}\}_{i=1}^{L_C}\right)$. We assume $\pi_C$ to be a fixed, capable LMM, and focus here on training the candidate selector $\pi_S$.

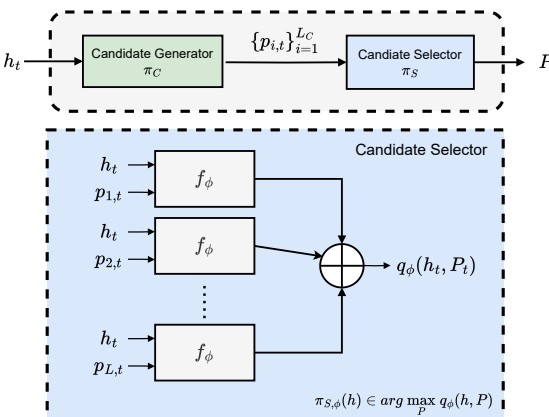

Figure 2: **Top**. PASTA policy framework: The LMM is used as a candidate generator of a candidate set, from which the candidate selector policy is used to select a slate. **Bottom.** Each prompt in the slate is evaluated individually using a prompt-value model, and the overall slate value is calculated as the average of the individual prompt values.

## 3.2. Value-Based Candidate Selector

We use a state-action value function to define our selector policy, $\pi_S$. Given candidates $\{p_i\}_{i=1}^{L_C}$ and a state-action value model $q_\phi(h, P)$ parameterized by $\phi$, we define the selector policy by

$$\pi_{S,\phi}(h) \in \arg \max_{P \in \mathcal{P}_L\left(\{p_i\}_{i=1}^{L_C}\right)} q_\phi(h, P), \qquad (1)$$

where $\mathcal{P}_L(X) = \{P \in X : |P| = L\}$ is the set of all possible slates of size $L$. Enumerating $\mathcal{P}_L\left(\{p_i\}_{i=1}^{L_C}\right)$ is, of course, computationally expensive when $L_C$ is large (a detriment to both training time and inference efficiency). Inspired by Ie et al. (2019), we decompose the value function into a (weighted) average of prompt-values $f_\phi : \mathcal{H} \times \mathcal{P} \mapsto \mathbb{R}$. Specifically, we compute the value of each prompt, and estimate the value of a slate by:

$$q_\phi(h, P) = \frac{1}{L} \sum_{p \in P} f_\phi(h, p).$$

This approximation reduces the exponential complexity of finding the best slate to $\mathcal{O}(L_C \log L_C)$ (i.e., by sorting the prompt values over the candidate set). See Figure 2 for a schematic of the PASTA policy and value function. PASTA first prompts an LMM $\pi_C$ (i.e., candidate generator) to generate $L_C$ candidates $\{p_i\}_{i=1}^{L_C} \sim \pi_C(h)$. It then uses its candidate selector $\pi_S$ to select a slate $P$ of prompts according to Equation (1).

We train PASTA using *implicit Q-Learning (IQL)* (Kostrikov et al., 2021), which estimates the TD error with the Bellman optimality operator. IQL employs a soft estimate with a value function trained to approximate the high expectile

without assessing state-actions outside the dataset distribution, and has proven effective in offline RL with LLMs (Snell et al.; Zhou et al., 2024). The IQL loss is:

$$\mathcal{L}(\phi, \psi) = \mathbb{E}_{h_t, P_t, r_t, h_{t+1} \sim \mathcal{D}}\Bigg[(q_\phi(h_t, P_t) - r_t$$
$$- v_\psi(h_{t+1}))^2 + L_2^\alpha(q_{\hat{\phi}}(h_t, P_t) - v_\psi(h_t))\Bigg],$$

where $L_2^\alpha(x) = \left|\alpha - 1_{\{x<0\}}\right| x^2$, $\alpha \in [0.5, 1]$ and $v_\psi$ is the $\alpha$-expectile value estimate. Most notably, the IQL loss does not require searching for the best slate (which is oftentimes out of the distribution of the training data), making it very efficient for training (see Appendix E.1 for further details).

## 4. PASTA Dataset

Training PASTA relies on the availability of *sequential* user-agent interaction data. While single-turn T2I preference data is readily available (Wu et al., 2023; Kirstain et al., 2023; Pressman et al., 2022; Liang et al., 2024), sequential datasets are not. Hence, we collect human-rater data *for our sequential setup*, and further enrich it with simulated data. Below, we describe our data creation process. All of our datasets are open-sourced here: https://www.kaggle.com/datasets/googleai/pasta-data.

### 4.1. Human Rater Sequential Data

We use human raters to gather sequential user preferences data for preference-adaptive T2I generation.[4] Participants are tasked with interacting with an LMM agent for five turns. Throughout our rater study we use a Gemini 1.5 Flash Model (Gemini-Team, 2024) as our base LMM, which acts as an agent. At each turn, the system presents 16 images, arranged in four columns ($L = 4, M = 4$), each representing a different prompt expansion derived from the user's initial prompt and prior interactions. Raters are shown only the generated images, not the prompt expansions themselves.

At session start, raters are instructed to provide an initial prompt of at most 12 words, encapsulating a specific visual concept. They are encouraged to provide descriptive prompts that avoid generic terms (e.g., "an ancient Egyptian temple with hieroglyphs"instead of "a temple"). At each turn, raters then select the column of images preferred most (see the UI used in Appendix B); they are instructed to select a column based on the quality of the *best* image in that column w.r.t. their original intent. Raters may optionally provide a free-text critique (up to 12 words) to guide subsequent prompt expansions, though most raters did not use this facility.

---

[4]Raters were paid contractors. They received their standard contracted wage, which is above the living wage in their country of employment.

After five turns, raters enter an evaluation phase where they answer questions about each turn, including: (1) re-confirmation of their preferred columns; (2) quantifying whether the image slate in the current turn is better than the previous; and (3) a free-text explanation for their selection. This process yields a dataset comprising sequential interaction trajectories with user preference feedback and provides valuable insight into the dynamics of multi-turn, preference-adaptive T2I generation. Our human rater dataset includes over 7,000 five-step sequences (more than 500,000 images total), annotated by about 100 human raters using a random candidate selector. See Appendix B for a comprehensive description of the rater study.

### 4.2. Simulated Sequential Data

Simulated data can drastically improve performance of RL agents (Kaiser et al., 2019; Yu et al., 2021; Liu et al., 2023; Hafner et al., 2020; Micheli et al., 2022). For this reason, we enrich the rater data above with additional *simulated* agent-user interaction data. To do so, we develop a *user model* that encodes distinct user *types* that reflect a range of preferences. This model has two components: (1) A *user choice model* which predicts the image column a user selects; and (2) a *user utility model* which predicts a user's satisfaction with an image slate. We outline details of the user model in Section 5.

We begin by sampling a user and initial prompt from a joint prior. We use a randomized exploration policy for the agent, encouraging diverse interactions to distinguish preferences across user types. At each turn in a simulated trajectory, the agent proposes a slate of prompt expansions, and the user (via the choice model) selects their preferred prompt. The user utility model is then used to assign a satisfaction score for the images generated using the selected prompt. This process is repeated for five turns (see Appendix C for further details and analysis of our data generation process). The simulated user data includes over 30,000 rollouts (exceeding 2.5 million images). Augmenting our human-rater data with this simulated data allows for more robust training of PASTA, as we show later in Section 6.

## 5. User Model

The data generation process described in Section 4.2 requires a user simulator consisting of both user choice and utility models. We leverage our sequential human-rater data (Section 4.1), together with large-scale open-source single-turn T2I preference data (Wu et al., 2023; Kirstain et al., 2023; Pressman et al., 2022), to build these models.

We first propose a user preference model based on fundamental *choice theory* (Pennock et al., 2000) that mimics user's preference behaviors in an interactive recommenda-

tion system. Specifically, this model takes as input a user type $u \in \mathcal{U}$, an initial prompt $p \in \mathcal{P}$, and a slate of images $P_t = \{I_{m,l,t}\}_{\ell \neq \ell, m=1}^{L,M}$ and estimates the probability that $u$ prefers image column $\{I_{m,\ell^*,t}\}_{m=1}^{M}$ over the others $\{I_{m,l,t}\}_{\ell \neq \ell^*, m=1}^{L,M}$, i.e.,

$$P\Big(\{I_{m,\ell^*,t}\}_{m=1}^{M} \succ \{I_{m,l,t}\}_{\ell \neq \ell^*, m=1}^{L,M} | u, p \Big). \quad (2)$$

In the typical *single-turn* setup (Wallace et al., 2024), by setting $M = 2$, $L = 1$, and not assuming any user types, Equation (2) reduces to a probabilistic model that estimates the "global" (non-user specific) pair-wise preference between two images, i.e.,

$$\mathbb{E}_{u \sim \nu_0}[\, P(I_1 \succ I_2) \,|\, u, p \,].$$

By contrast, our model is able to capture user-specific preferences over a generic sets of multiple images.

Our user choice model $C$ exploits the preference model in Equation (2), making the usual assumption that users select images based on their inherent preferences; i.e.,

**Assumption 5.1** (Choice-Preference Equivalence). Assume $C \equiv P$, where $P$ is the preference model in Equation (2).

While, in general, user choice may also be be biased by other factors (e.g., previous interactions, position bias, etc.), this idealized model proves very useful in our experiments.

To further model user utility, motivated by the axioms of expected utility theory (Von Neumann & Morgenstern, 2007), we model user utility $R : \mathcal{U} \times \mathcal{P} \times \mathcal{I}^{ML} \mapsto \mathbb{R}$, leveraging the preference model in Equation (2) to ensure a user's utility is consistent with their preferences. Specifically, we assume that a user prefers one set of images to another if the utility of the former images is higher, as formalized by the following assumption.

**Assumption 5.2** (Utility Consistency). For any $u, p$, Equation (2) is satisfied for $\ell^* \in \{1, \ldots, L\}$ iff for all $\ell \neq \ell^*$

$$R\Big(\{I_{\ell^*,m,t}\}_{m=1}^{M} | u, p\Big) > R\Big(\{I_{\ell,m,t}\}_{m=1}^{M} | u, p\Big).$$

Assumptions 5.1 and 5.2 ensure a tight coupling of our user utility and choice models, which we leverage in model training.

### 5.1. A Parametric User Model

To train our user utility and choice models, we employ a parameterized score model $s_\theta : \mathcal{U} \times \mathcal{I} \times \mathcal{P} \mapsto [0, 1]$ to serve as the backbone of our user simulator. We assume a discrete set of $K$ (unknown) user types, i.e., $\mathcal{U} = \{1, 2, \ldots, K\}$.

Exploiting Assumptions 5.1 and 5.2, we define the utility of a user type $k$ over an arbitrary image slate $\{I_{m,\ell,t}\}_{m=1}^{M}$ that

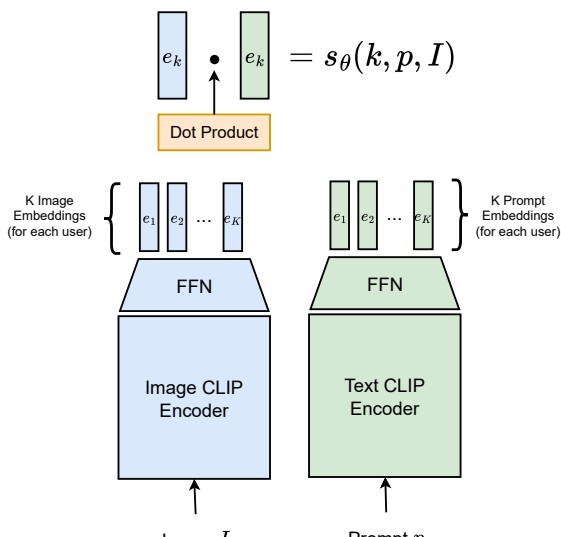

Figure 3: Score function architecture. The image and prompt are fed into CLIP encoders followed by user encoders to generate $K$ user embeddings. The score for each user type is the inner product between the corresponding user image embedding and user text embedding.

associates to a prompt $p$ using the following model:

$$
\begin{aligned}
R_{\ell,t}^{k,\theta} &:= R_\theta\Big(\{I_{m,\ell,t}\}_{m=1}^M | u = k, p\Big) \\
&= \mathrm{Agg}(s_\theta(k,p,I_{1,\ell,t}), \ldots, s_\theta(k,p,I_{M,\ell,t})),
\end{aligned}
\tag{3}
$$

where Agg is an aggregator (e.g., average, max, Softmax sampler) operator. User choice probabilities are given by

$$
\begin{aligned}
&C_\theta(k,p, \{I_{m,1,t}\}_{m=1}^M, \ldots, \{I_{m,L,t}\}_{m=1}^M) \\
&= \mathrm{Softmax}\Big(\tau_\theta R_{1,t}^{k,\theta}, \ldots, \tau_\theta R_{L,t}^{k,\theta}\Big),
\end{aligned}
\tag{4}
$$

where $\tau_\theta = h_\theta\Big(R_{1,t}^{k,\theta}, \ldots, R_{L,t}^{k,\theta}\Big)$ is a parameterized temperature constant which depends on the scores of different image columns. Such a temperature parameterization ensures our user choice model satisfies Assumption 5.2, while allowing for greater flexibility in modeling.

To balance model capacity and computational efficiency, our score function adopts pre-trained CLIP (Radford et al., 2021) text and image encoders, followed by user encoders (a head for each user type). The user encoder transforms CLIP embeddings into a user-type-specific representation that captures individual preferences for images and prompts. The final score of an image-prompt pair is the inner product of the (user-type-specific) image embedding and corresponding text embedding, multiplied by a learned temperature parameter (see a schematic of our architecture in Figure 3).

We employ an Expectation-Maximization (EM) framework to learn a user model capable of capturing diverse pref-

erences. This model leverages the score function $s_\theta$ that assesses the compatibility between image-prompt pairs and different user types. The EM algorithm iteratively refines the model parameters $\theta$ and a user type prior distribution $\eta$ to maximize the likelihood of observed user feedback. Training proceeds in two phases: a main training phase on large-scale datasets with frozen CLIP parameters, followed by a fine-tuning phase on human-rated data where all parameters are trained. We detail the training procedure, specific loss functions tailored to different data types (single-turn preferences, relevance scores, and sequential multi-turn), the model architecture, and hyperparameter settings in Appendix D.1.

# 6. Experiments

To assess the effectiveness of our approach, we conduct two empirical evaluations. First, we evaluate the quality of our user model and the simulated data it generated. This involves analyzing the model's ability to accurately capture user preferences and generate realistic interaction data. Second, we study the performance of PASTA, our multi-turn T2I agent, trained using our rater and simulated datasets. We perform a comprehensive analysis at each stage to assess the quality of the user model, the accuracy of the simulated data, and the overall performance of the T2I agent.

## 6.1. Setup

The slate size, number of images per prompt, and problem horizon are fixed $L = 4$, $M = 4$ and $H = 5$ in all experiments. We use Stable Diffusion XL (Podell et al., 2023) as our T2I model and multimodal Gemini 1.5 Flash (Gemini-Team, 2024) as our (prompt) candidate generator LMM, with $L_C = 25$. Our learned value function is fine-tuned from Gemma 2B (Gemma-Team, 2024). Our experiments use a sparse reward, with rewards provided only in the final round, as our primary interest lies in the end result of interactive refinement process. We use softmax sampling for the user model's aggregation function in Equation (3), as raters were instructed to select a column based on the best image from that column.

To further encourage diversity in the generation of prompt candidates, we require the LMM prompt generator to partition the candidate set into five categories, with each category focusing on a different aspect of the prompt expansion. Categories include both generic forms (e.g., rephrasing, adding more details) as well as prompt-specific approaches (e.g., different types of dogs, cloud shapes, etc.). Moreover, we restrict the candidate selector to select at most one prompt from each category. This simple technique induces significant diversity (hence implicit exploration), and is used for data generation, policy training, and at inference time.

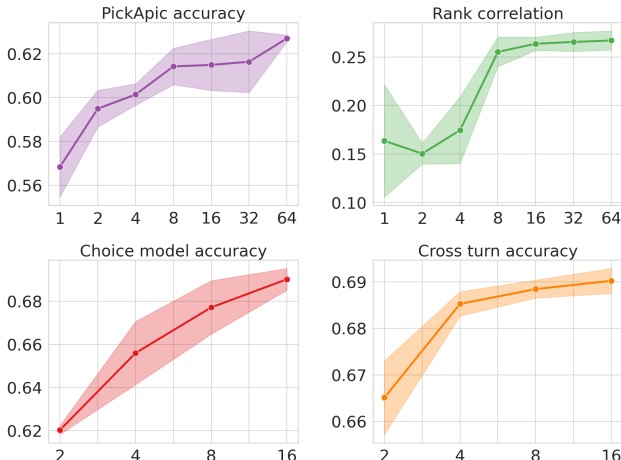

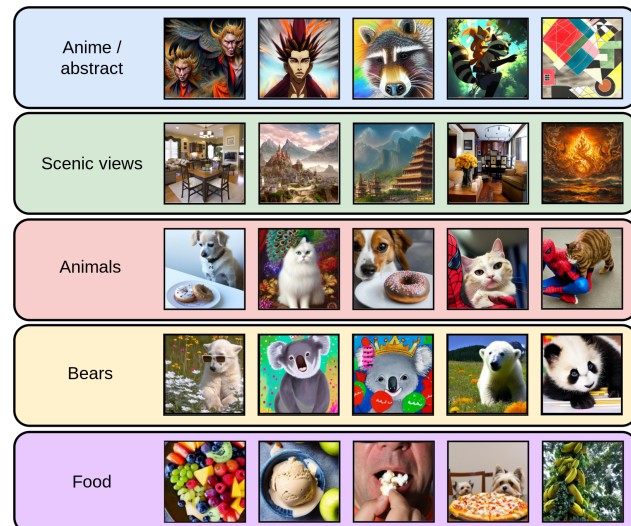

Figure 4: The graphs present the performance of a trained user model as a function of the number of user types considered. **The top row** displays the model's accuracy on the Pick-a-Pic test set (left) and its Spearman's rank correlation on the HPS test set (right). **The bottom row** shows the model's choice accuracy (left) and cross-turn preference accuracy (right), both evaluated on our human-rated test set.

Figure 5: **Emergence of user-specific preferences:** Each row displays the top five images—scored by the user model, prior to fine-tuning—from the HPS test set for one of five specific user types. We highlight user types where differences across the top five images were especially salient. The category labels (Animals, Food, etc.) are simply meant to be evocative on the style or content of the most preferred images.

### 6.2. User Model Evaluation

We evaluate the quality of our user model by assessing its ability to capture user preferences and generate realistic interaction data. We train the model using large-scale single-turn datasets (HPS V2 (Wu et al., 2023), Pick-a-Pic (Kirstain et al., 2023), and Simulacra Aesthetic Captions (Pressman et al., 2022)) and fine-tune it with our human-rater sequential data. During training, CLIP model weights are initially frozen, focusing on learning user image and text encoder parameters. See Appendix D for full training details, including loss functions and architectures.

We analyze model performance in two stages. First, we assess prediction accuracy of the and Pick-a-Pic testset (Kirstain et al., 2023) and ranking using Spearman's rank correlation (Spearman, 1961) on the HPS dataset (Wu et al., 2023). Second, we evaluate prompt choice prediction accuracy and cross-turn preference accuracy on our human-rated data. Results in Figure 4 indicate that increasing the number of user types significantly improves performance, plateauing around 16 types. This highlights the importance of modeling diverse user preferences. Notably, even with a moderate number of user types, our model achieves approximately 70% accuracy on the human-rated dataset metrics. A further notable observation is the emergence of visually distinct domain preferences for different user types; this tends to become more prominent as the number of user types increases. An illustration for the 32-type model is shown in Figure 5.

### 6.3. PASTA Results

Using our user model, we generate more than 30,000 simulated user-agent trajectories. We also generate reward labels for our human-rater data using the user model, and train our state-action value function in three different settings: using only human rater data, using only simulated data, and using both data sources. The T2I agents trained by these algorithms are evaluated with human raters, for which the raters judge whether the current turn's chosen image was better, worse, or equally preferred to that in the previous turn. To make PASTA more efficient for human-rater studies, we distill the T2I agent into a single, fine-tuned Gemini 1.5 Flash (Gemini-Team, 2024) LMM, and serve that in real-time to generate the proposed image slates directly (without explicit prompt expansions). We also conduct an experiment with simulated users, using our user model, whose results are described in Appendix F together with other experiments for both PASTA and the user model.

We compare our agents (each using a different state-action value function) with an untrained multi-modal Gemini 1.5 Flash model. The results of human-rater evaluations are shown in Figure 6. These results demonstrate that training on real human data enhances our agents' performance over the pre-trained LMM baseline (i.e., over the off-the shelf Gemini 1.5 Flash), whereas training with synthetic data alone leads to a slight decline in performance. Unsurprisingly, training on both real and synthetic datasets offers the best performance. This suggests that, while our user simulation may not fully reflect the real-world distribution of user behaviors (partially due to its simplified assumptions), the additional data generated by our user model does capture some key dynamics of real user interactions, yielding more accurate user choice predictions and sequentially generated

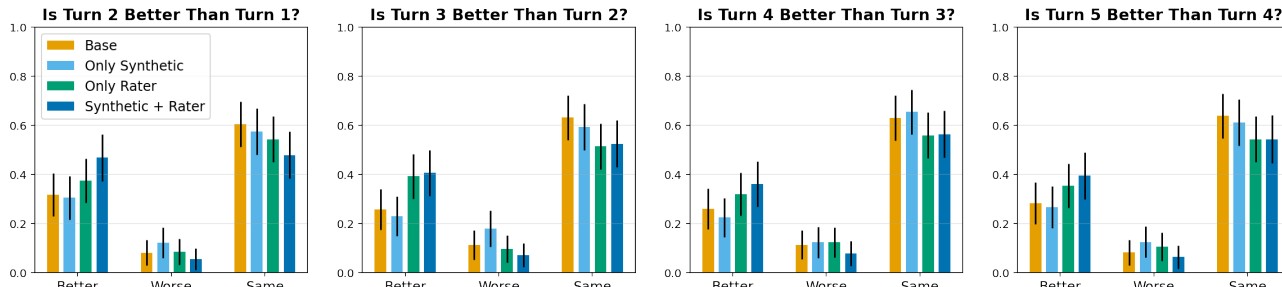

Figure 6: **PASTA Evaluation.** We ask raters to determine, for each interaction turn, whether the shown images are an improvement over the previous turn. The raters can choose one of three options: Better, Worse, or Same. Results show percentage of raters who chose each option. Experiments compare the base Gemini Flash model (without fine-tuning) to PASTA (also using Gemini Flash) varied by training on different datasets: (1) only simulated user data, (2) only real rater data, and (3) the combination of both datasets. Plots show 99% confidence intervals.

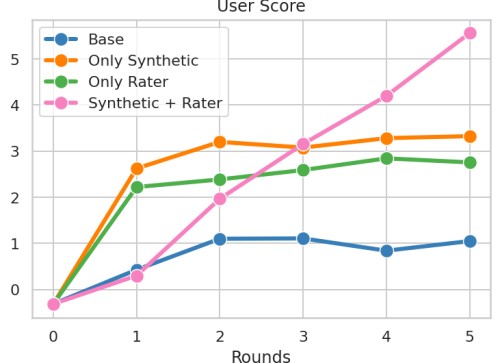

Figure 7: **PASTA Evaluation with the Score Model.** We use our trained user model to evaluate the rollouts of real users with different baselines.

images that are tailored to user preferences. Moreover, a high rank correlation between real user preferences and the predictions from our learned model over a vast number of user types further supports this conclusion. We also present the user model score for each step averaged over all rollouts in Figure 7. The user type was chosen as the one that maximizes the posterior over the user's data. The results show that PASTA monotonically improves the user score along the rollouts and are consistent with the real users' results in Figure 6. This evaluation further validates the correlation between our user model and real users.

Finally, we conduct new human rater evaluations over the final turn of PASTA. Particularly, we conduct a rater study where we explicitly show raters the final turn images (for a given prompt) of PASTA compared to the baseline Gemini 1.5 Flash model and ask them to compare the generated last turn images. We find that PASTA receives an 85% relative improvement over the baseline model when directly comparing the last turn. We believe that directly comparing the last turn emphasizes the strength of our approach and the significance of our results in terms of the outcome of using our model.

### 6.4. Generalization: Flux T2I model

We use Flux.1 [5] together with our PASTA agent and compare it to the baseline Gemini 1.5 Flash. Particularly, we do not retrain PASTA over new Flux.1 data, but rather test PASTA's capability to generalize from the SDXL T2I model to the Flux.1 T2I model. We present the raters the final images constructed by PASTA against the final images generated with the Gemini baseline. Indeed, our results show that PASTA achieves a 20% relative improvement over the baseline without any further training over Flux data. This highlights PASTA's ability to adapt to different T2I models.

### 6.5. Abstract Prompts with Simulated Users

To visualize the differences in user preferences and PASTA's ability to adapt to them, we use PASTA with various user types using broad, abstract prompts such as "an image of happiness". Figure 8 shows an example of different rollouts of three distinct user types. We observe a clear preference emerging, favoring specific styles or content. All users starts with the same prompt and initial images. We present only the images and their corresponding prompt-expansion at the last 5-th step. Each color represent a different user type. For more examples, see Appendix F.2.

## 7. Related Work

Our work is related to recent advances in interactive and preference-adaptive image generation. Existing methods explore iterative refinement using visual feedback (von Rütte et al., 2023; Wang, 2024) or leverage LLMs for prompt engineering (Liu et al., 2024; Brade et al., 2023; Hao et al., 2024), often focusing on optimization of perceptual quality or user satisfaction in a single interaction (Fan et al., 2023; Chidambaram et al., 2024; Wallace et al., 2024; Wu et al., 2023). Our PASTA framework extends this by framing multi-turn image generation as a sequential decision-making

---

[5]https://flux1.ai/

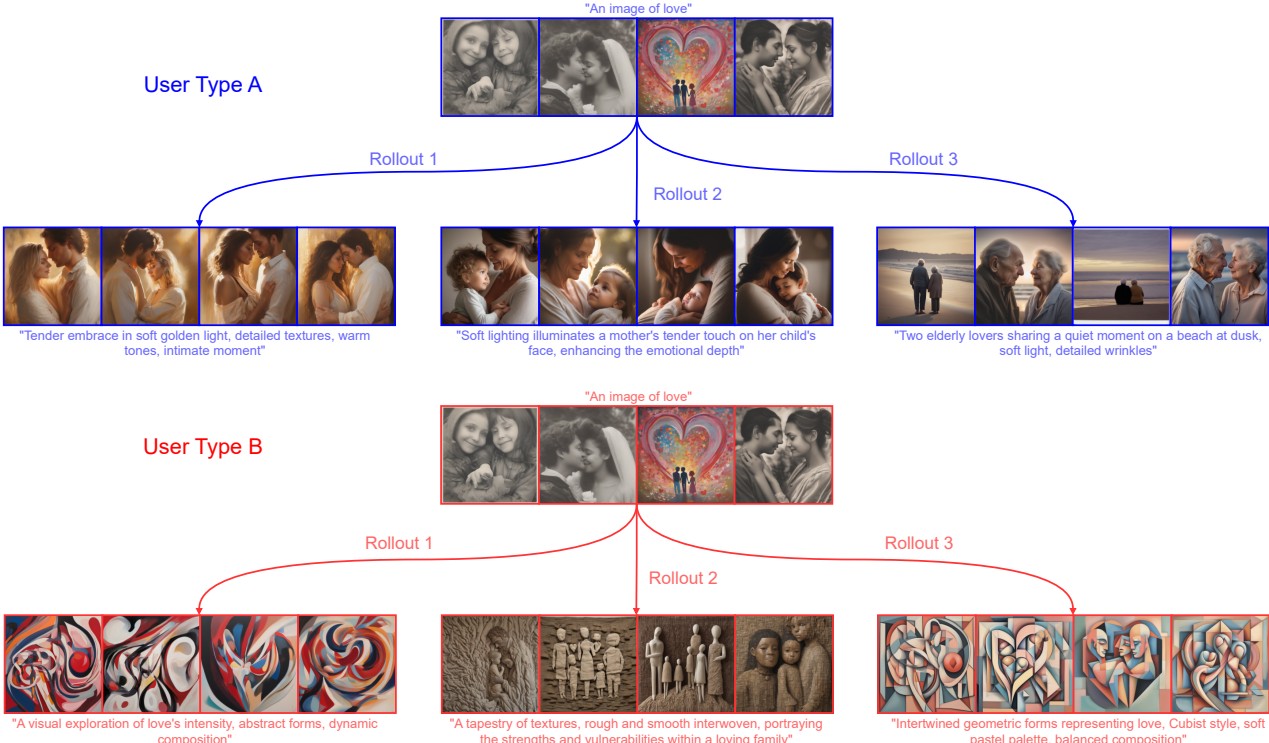

Figure 8: An illustration of three different rollouts of PASTA with two different user types for the abstract initial prompt: "An image of love". Rollouts show the final images in the rollout (i.e., at step $t = 5$).

problem, where an RL agent learns to extend and personalize a prompt through a series of expansions guided by user feedback. This enables iterative refinement towards a desired visual outcome, capturing nuances in user preferences across diverse user types. Notably, while many existing methods focus on modifying internal aspects of text-to-image (T2I) models (e.g., diffusion noise (Tang et al., 2023), attention maps (von Rütte et al., 2023) or prompt embeddings (Salehi et al., 2025)), PASTA operates by directly adapting the input prompt, offering a model-agnostic approach to adaptive preference learning in T2I generation.

PASTA draws upon a rich history of preference elicitation (PE) research (Boutilier, 2002; Chajewska et al., 2000; Keeney, 1993; Salo & Hamalainen, 2001), especially that in content-based recommender systems (Boutilier et al., 2003; Spearman, 1961; Rashid et al., 2002; Meshi et al., 2023), and addresses the challenge of optimizing PE for diverse downstream tasks. Existing research highlights the importance of diversity in recommendations (Sidana et al., 2018) and explores diversity in elicitation strategies (Parapar & Radlinski, 2021). PASTA tackles this by training an agent that considers both immediate user feedback and long-term goals to generate preference-aligned images. Our novel data collection and simulation methodology further support the training of robust user simulators, which we show to greatly improve policy learning.

## 8. Conclusion

This work introduced PASTA, a novel RL agent for preference-adaptive and sequential T2I generation. We formulated the problem of iterative prompt expansions with LMMs and T2I models as a sequential decision-making problem that drives a collaborative, multi-turn image generation with the user. Critically, we have produced a new dataset that captures *sequential* interactions between an agent and a user, which we release to support further investigation of new T2I techniques in the research community. We developed user utility and choice models, learned from this dataset, which we used to create a user simulator that enabled generation of additional synthetic data (which we release as well). Finally, we used our sequential data to train PASTA, and demonstrated its effectiveness through comprehensive human evaluations, showing significant improvements in user satisfaction.

## Impact Statement

This paper presents work whose goal is to advance the field of Machine Learning. There are many potential societal consequences of our work, none which we feel must be specifically highlighted here.

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

# A. Latent Markov Decision Process (LMDP)

We begin with defining *Latent Markov Decision Process* introduced in (Hallak et al., 2015; Kwon et al., 2021).

**Definition A.1.** Suppose that a set of MDPs $\mathcal{M}$ with a joint state space $\mathcal{S}$ and joint action space $\mathcal{A}$ in finite horizon setting with horizon $H$. let $K = |\mathcal{M}|$, $S = |\mathcal{S}|$ and $A = |\mathcal{A}|$. Each MDP $\mathcal{M}_k \in \mathcal{M}$ it a tuple $(\mathcal{S}, \mathcal{A}, T_k, R_k, \nu_k)$ where $T_k : \mathcal{S} \times \mathcal{A} \to \Delta_{\mathcal{S}}$ is a transition probability that maps a state-action pair into a distribution over the next state, $R_k : \mathcal{S} \times \mathcal{A} \mapsto \Delta_{[0,1]}$ a probability measure for rewards that maps a state-action pair into a distribution over rewards and initial state distribution $\nu_k$. Let $\eta_1, \eta_2, \ldots, \eta_K$ be the mixing weights of LMDPs such that at the start of every episode, one MDP $\mathcal{M}_k \in \mathcal{M}$ is randomly chosen with probability $\eta_k$.

The goal of the problem is to find a policy $\pi$ within a policy class $\Pi$ that maximizes the expected return:

$$v^*_{\mathcal{M}} = \max_{\pi \in \Pi} \sum_{k=1}^{K} \eta_k \mathbb{E}_k^{\pi} \left[ \sum_{t=1}^{H} r_t \right],$$

where $\mathbb{E}_k^{\pi}[\cdot]$ is expectation taken over the $k$-th MDP with policy $\pi$. The policy $\pi : \mathcal{H} \times \mathcal{S} \mapsto \Delta_{\mathcal{A}}$ maps the current state and history into distribution over actions. Generally, the history is all experience seen so far $\mathcal{H} = (\mathcal{S}, \mathcal{A}, [0, 1])^*$. When the model parameters are known, history is a sufficient statistics and can be summarized into belief states:

$$b_1(k) = \frac{\eta_k \nu_k(s_1)}{\sum_k \eta_k \nu_k(s_1)},$$
$$b_{t+1}(k) = \frac{b_t(k) T_k(s_{t+1}|s_t, a_t) R_k(r_t|s_t, a_t)}{\sum_k b_t(k) T_k(s_{t+1}|s_t, a_t) R_k(r_t|s_t, a_t)}.$$

We formulate our reinforcement learning problem as a latent MDP, where each user from a discrete set induces distinct transition and reward kernels. Specifically, in our sequential text-to-image decision problem the state space is a prompt and image slate $\mathcal{S} \equiv \mathcal{P} \times \mathcal{I}^{ML}$ and the action space is the prompt slate $\mathcal{A} \equiv \mathcal{P}^L$. The transition kernel for the $k$-th user type is composed of the T2I model and user choice model, i.e. the next state is the tuple composed of slate of images generated from the chosen slate (the action) and chosen prompt:

$$s_{t+1} = \left( \left\{ \{I_{m,\ell,t}\}_{m=1}^{M} \right\}_{\ell=1}^{L}, p_{c_t} \right),$$

where $I_{m,\ell,t} \sim G(p_{\ell,t})$ and $c_t \sim C(k, p_0, I_{1,1,t}, \ldots, I_{M,L,t})$. The reward kernel corresponds to the user's utility function:

$$r_t \sim R(k, p_0, I_{1,\ell_t,t}, \ldots, I_{M,\ell_t,t}).$$

Similar to standard LMDPs, the posterior update during user model training in **??** mirrors the belief state computation, driven by observed samples collected from diverse human raters.

# B. Human Rater Dataset

To gather data for training and evaluating PASTA, we conducted a human rater study. We recruited paid contractors as raters, and utilized a web-based interface where raters interacted with a Gemini 1.5 Flash Model acting as the agent in our multi-turn image generation process.

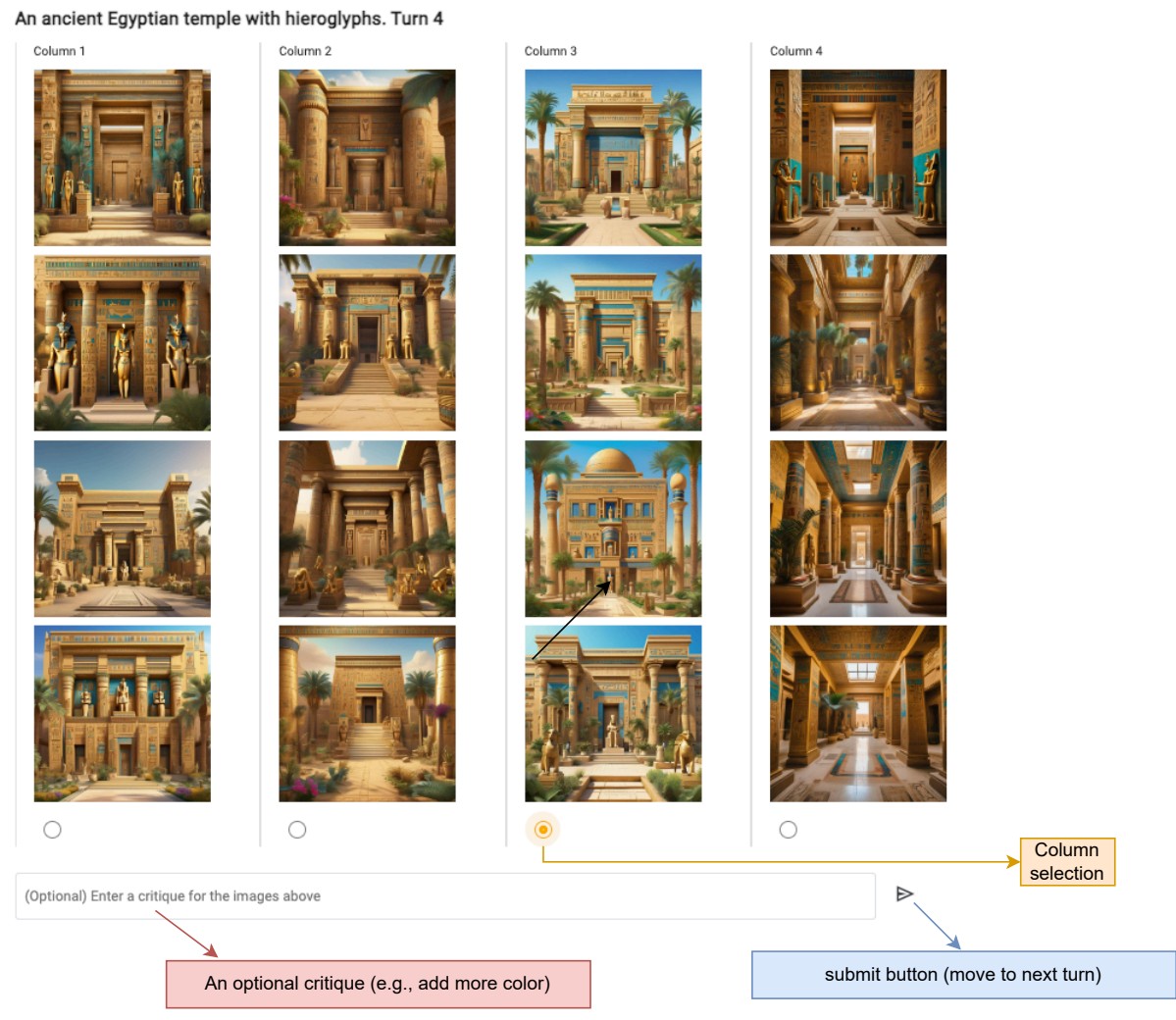

Figure 9: A screenshot of the interface raters used in the interaction phase. Raters were not shown the prompt expansions. They were instructed to choose a preferred column (based on the best image of each column). Raters could optionally also add an up to 12 word critique.

## B.1. Rater Instructions: Interaction Phase

Each rater participated in a five-turn interaction session. At the beginning of the session, raters were presented with the task's instructions. They were then asked to provide an initial text prompt, with a maximum length of 12 words, representing a specific visual concept they wanted to see realized as an image. The guidelines stressed that prompts should encapsulate meaningful intent (thinking about a specific image they wished to see) and avoid generic prompts (e.g., "a temple"). Instead, they were encouraged to use descriptive prompts within the 12-word limit (e.g., "An ancient Egyptian temple with hieroglyphs"). They were assured that they need not worry about fitting all of their thoughts into the 12-word limit, as the interaction process with the agent would help them refine their concept.

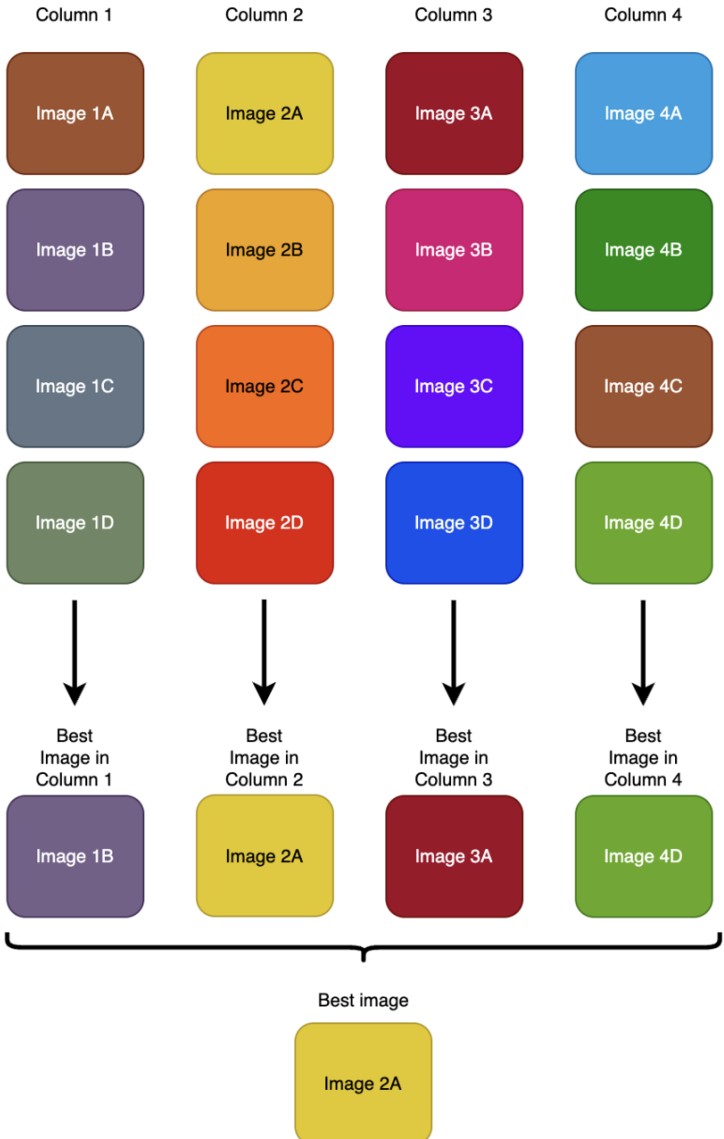

Figure 10: Illustration shown to raters on how to select a column. Columns should be compared based on the best image of each column.

In each of the subsequent five turns, the LMM agent presented the rater with 16 images, arranged in four columns (L=4, M=4). Each column represented a different prompt expansion derived from the rater's initial prompt and their choices in previous turns. Figure 9 illustrates this setup. Critically, the raters were only shown the generated images, not the expanded prompts themselves. This ensured their feedback was based purely on visual perception and alignment with their initial intent, rather than on their interpretation of the textual modifications made by the agent.

The core task for the rater in each turn was to select the column of images they preferred most. The instructions, visually depicted in Figure 10, emphasized that this selection should be based on the best image within each column, judged in terms of general quality and its correspondence with the rater's original prompt intent. For example, if Image 2A was the best overall image of the 16, then Column 2 should be selected. This selection strategy was used to help reduce burden on raters. It also benefited our choice model training procedure, which used this as inductive bias for training our model.

Optionally, raters could provide a free-text critique of the presented images, limited to 12 words (Figure 9). This critique could address any aspect of the images, such as style, composition, color, or content, but raters were explicitly instructed not

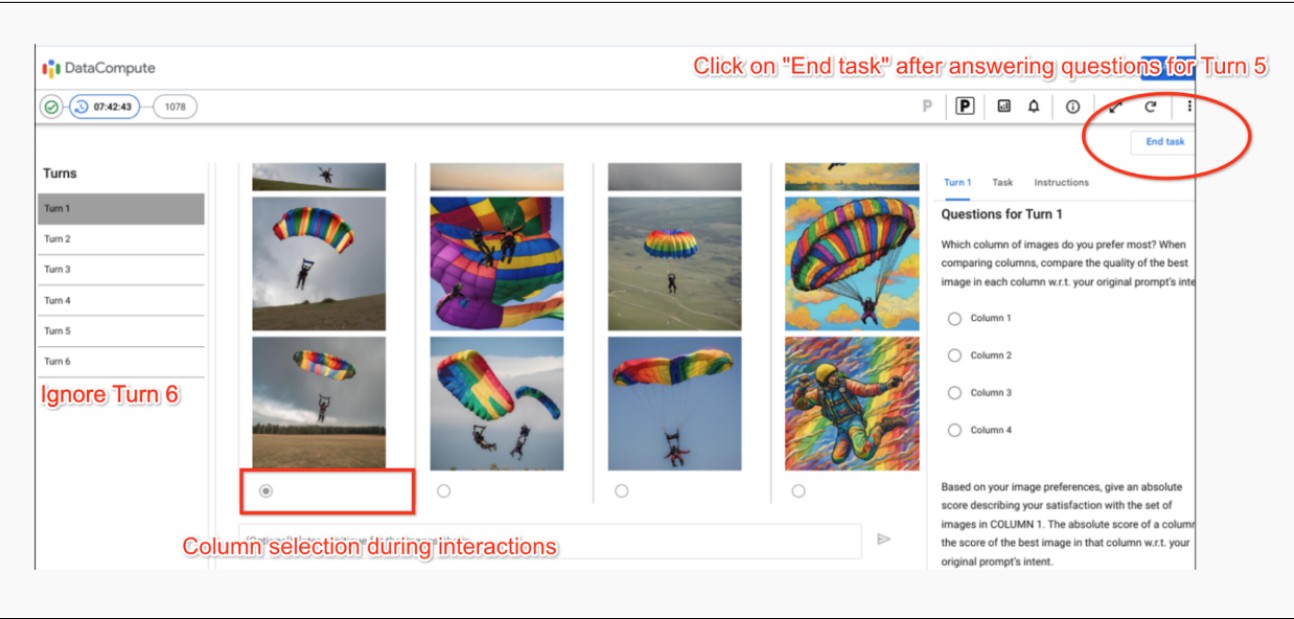

Figure 11: Screenshot of the evaluation phase for raters. This second phase (after the interaction phase) included questions about every turn. See more details in Appendix B.2.

to use the critique field to change their original intent. For example, "Add more color" is a valid critique. The purpose of the critique was to provide additional signals to the agent for refinement within the scope of the original visual concept, not to redirect the generation process towards a different image altogether. Overall, approximately 20% of the collected data have one or more critiques.

### B.2. Rater Instructions: Evaluation Phase

After completing five interaction turns, raters were prompted to click the "Enter rater mode" button. The system then transitioned to the evaluation phase. For each of the five turns, raters answered a set of questions designed to capture detailed feedback on their experience and the agent's performance. Figure 11 demonstrates the interface used during this stage. For each turn, raters were asked the following questions, with their chosen column highlighted for reference:

1. Column Preference: Raters were asked to re-confirm their preferred column for each turn, ensuring consistency in their choices (See Figure 11).

2. Absolute Column Satisfaction: For each of the four columns in each turn, raters provided an absolute satisfaction score on a 5-point scale (1: very dissatisfied, 5: very satisfied). This score reflected their satisfaction with the best image in the column, relative to their original prompt intent. This absolute scoring allowed for a finer-grained assessment of the individual prompt expansions, capturing nuances beyond the simple relative ranking implied by column selection.

3. Selection Rationale: For their selected column in each turn, raters were asked to provide a free-text explanation of their choice.

4. Turn-over-Turn Improvement: For turns 2-5, raters assessed whether the set of images in the current turn was better, worse, or about the same as the set of images in the previous turn. This comparison, again focused on the quality of the selected columns in each turn relative to the original prompt intent, provided a measure of the agent's progress and its ability to refine the generated images over time.

5. Overall Turn Satisfaction: Raters provided an overall absolute satisfaction score (1-5 scale) for all of the images presented in each turn. This average score across columns gave a holistic impression of the agent's performance in that turn, complementing the column-specific feedback.

Raters were finally instructed to press the "End Task" button (as shown in Figure 11) to complete their session, which submitted their responses for all five turns.

### B.3. Limitations of Rater Study

A recognized limitation of the rater study conducted in this work is the potential bias between groups of raters. In Figure 6 we compare PASTA trained on different datasets with the pre-trained Gemini Flash model. We could not concurrently run different experiment arms because of a lack of A/B testing infrastructure which shows different agents to raters at random. Consequently, the timing of experiment arms and rater availability could have introduced bias. Furthermore, the rater pool might not be fully representative of all potential user demographics for PASTA, and the initial prompts provided to raters might not have sufficient diversity.

Human errors and system issues, such as connection timeouts, were observed during the study. We did not implement any formal mechanisms for raters to report errors during specific sessions although raters can somehow "correct" their mistakes during the evaluation phase. Raters could prematurely abandon sessions without recording data from problematic sessions. Raters could provide a critique of the presented images but subsequent generations might not be aligned with these critiques. Although raters were explicitly instructed not to evaluate based on critique alignment, it might still cause worse rater experience for those who provided critiques.

### B.4. Generating Rewards for Human Rater Data with User Model

To leverage human rater data for value model training, we utilize our learned user utility function to generate rewards for each interaction step. For each trajectory, we first compute the posterior distribution over user types using the model described in Appendix D.3. We then employ the utility function of the most likely user type to generate rewards for each timestep in the trajectory.

# C. Offline Simulated Sequential Data

This section describes the generation of an offline dataset for training PASTA. We detail the process of creating interaction trajectories with simulated users, based on our trained user model and a random prompt selection policy. We also address the challenge of MDP ambiguity inherent in offline learning, specifically in the context of diverse user types, and present conditions for ensuring identifiability of the offline dataset.

## C.1. Offline Dataset Creation

Leveraging our trained user model and user type prior, we generated a dataset comprising over 30,000 simulated user interaction trajectories. We assume independence between the initial prompt distribution and the user type distribution, i.e., $\nu(p_0, u) = \nu(p_0)\eta_0(u)$. Therefore, at the beginning of each trajectory, we approximate the true user prior with the learned prior from user model training ($\eta_0 \approx \eta$) and sample initial prompts uniformly from the available datasets, while limiting the initial prompts to be with no more than 10 words in order to avoid very large prompt at the end of the rollout.

To ensure diverse and representative data for training, the data generation policy needs to produce coherent prompts that encourage exploration and sufficiently cover the state-action space, thereby mitigating potential MDP ambiguity (discussed further in the next subsection). We achieve this by using the candidate generator detailed in Appendix E.2 coupled with a random selection policy, denoted as $\pi_{\text{rand}}$. This policy is constrained to select a maximum of one prompt per category, promoting exploration across different prompt categories.

At each interaction step, the candidate generator produces a categorized set of $L_C$ prompts. The random policy $\pi_{\text{rand}}$ then samples a slate of $L$ prompts, for which the text-to-image model generates $M$ images each. A prompt $c_t$ is selected according to the user choice model, and the corresponding user satisfaction level (reward) is recorded based on the utility function defined in Section 2.2.

## C.2. Identifiable Offline Dataset

After constructing the user choice model and utility function, we use a user simulator to generate an offline dataset for PASTA training. The offline reinforcement learning framework is chosen due to its computational efficiency during training. However, offline LMDP introduces the challenge of Markov Decision Process (MDP) ambiguity (Dorfman et al., 2021), and in our context, user type ambiguity:

**Definition C.1. User type ambiguity (Dorfman et al., 2021).** Consider data generated from a set of $K$ user types $\mathcal{U} = [1, 2, \ldots, K]$, each corresponding to a user choice model and utility function $\{C_i, R_i\}_{i=1}^K$. The data is collected by sampling a subset of $N$ user types $\{u_i\}_{i=1}^N \subset \mathcal{U}$, and applying their respective behavior policies $\{\pi_i^\beta\}_{i=1}^N$. This results in $N$ data distributions $\{\rho_{u_i, \pi_i^\beta}\}_{i=1}^N$ over image slates, initial prompt and rewards, with $\rho : \mathcal{P} \times \mathcal{I}^{ML} \times [0, 1] \mapsto [0, 1]$. We define the collected data as ambiguous if there exists a user type $u^* \in \mathcal{U}$ and two policies $\pi, \pi'$ such that $\rho_{u_i, \pi_i^\beta} = \rho_{u^*, \pi}$ and $\rho_{u_j, \pi_j^\beta} = \rho_{u^*, \pi'}$ for some $i \neq j$. Otherwise, the data is considered identifiable.

In essence, the data is ambiguous if a user type under different policies results in same data distributions of two different user types and their corresponding behavioral policies, regardless of how much data is collected. By introducing the concepts of identifying state-action pairs and overlapping state-actions, we can define a sufficient condition for identifiability.

**Definition C.2. Identifying Slate of Images.** For a pair of user types $i, j \in \mathcal{U}$, we define $(p_0, \{I_{m,\ell}\}_{m=1,\ell=1}^{M,L})$ as an identifying slate of images if $R(i, p_0, \{I_{m,\ell}\}_{m=1,\ell=1}^{M,L}) \neq R(j, p_0\{I_{m,\ell}\}_{m=1,\ell=1}^{M,L})$ or $C(i, p_0, \{I_{m,\ell}\}_{m=1,\ell=1}^{M,L}) \neq C(j, p_0, \{I_{m,\ell}\}_{m=1,\ell=1}^{M,L})$.

**Definition C.3. Overlapping Slate of Images.** Given data collected as described in Definition C.1. a slate of images $(p_0, \{I_{m,\ell}\}_{m=1,\ell=1}^{M,L})$ is said to overlap for a pair of user types $i, j \in \mathcal{U}$ if it has a positive slate-probability under both user types, i.e., $\rho_{i,\pi_i^\beta}(p_0, \{I_{m,\ell}\}_{m=1,\ell=1}^{M,L}) > 0$ and $\rho_{j,\pi_j^\beta}(p_0, \{I_{m,\ell}\}_{m=1,\ell=1}^{M,L}) > 0$.

A sufficient condition for identifiability is as follows:

**Proposition C.4.** *(Dorfman et al., 2021). Consider a data collected according to the settings in Definition C.1. The data is identifiable if, for every pair of distinct user types $i \neq j$, there exists an identifying slate of images that overlaps.*

Given that in our scenario user types vary not only in their utility functions but also in their choice models, the candidate

generator combined with the random sampler $\pi_{rand}$ is likely to satisfy the conditions in Proposition C.4. This approach introduces sufficient randomness to explore a wide range of states, allowing us to reach identifying slates of images for all user types, while still maintaining a coherent policy capable of generating high-quality prompts. In the rare case where two user types generate identical data distributions under $\pi_{rand}$, we consider them the same user type and merge them.

# D. User Model

We employ an Expectation-Maximization (EM) framework to learn a user model capable of capturing diverse preferences. This model leverages a score function $s_\theta$ that assesses the compatibility between image-prompt pairs and different user types. The EM algorithm iteratively refines the model parameters $\theta$ and a user type prior distribution $\eta$ to maximize the likelihood of observed user feedback. Training proceeds in two phases: a main training phase on large-scale datasets with frozen CLIP parameters, followed by a fine-tuning phase on human-rated data where all parameters are trained. We detail the training procedure, specific loss functions tailored to different data types (single-turn preferences, relevance scores, and novel sequential multi-turn data), the model architecture, and hyperparameter settings in the following subsections.

## D.1. Training

The user model, which always incorporates the score function $s_\theta$, is adapted to suit different datasets and label types. Specific model formulations and loss functions are detailed in subsequent subsections. $\mathcal{D}$ and estimate the log-likelihood loss as follows:

1. **Mini-Batch Sampling:** A mini-batch $\mathcal{B}$ is sampled from the dataset, and the posterior is estimated solely for samples within $\mathcal{B}$ using a target network to prevent overestimation.

2. **Prior Update:** An exponential moving average of the mini-batch posteriors is used to update the prior.

3. **Gradient Descent:** Instead of fully optimizing the model parameters, a single gradient descent step is taken with respect to the loss function:

$$\mathcal{L}(\theta) = -\mathbb{E}_{i\sim\mathcal{B},k\sim\gamma_i}[\log\sigma_\theta(x_i,y_i,k)].$$

The user model is always composed of the score function $s_\theta$ and may vary for different types of datasets and labels, and need to be adjusted accordingly. We provide detailed models and loss function at the following sub-sections. Algorithm 1 provides a complete description of the mini-batch training procedure. Training proceeds in two phases:

1. **Main Training:** The model is trained on a combination of large-scale datasets: HPS V2 (Wu et al., 2023), Pick-a-Pic (Kirstain et al., 2023), and Simulacra Aesthetic Captions (SAC) (Pressman et al., 2022). The CLIP model parameters are frozen during this phase.

2. **Fine-tuning with Sequential Data:** A shorter fine-tuning phase follows, utilizing human-rated data. Here, the entire model, including the CLIP model, is trained.

---

**Algorithm 1** Mini-Batch Expectation-Maximization User Model Optimization

---

1: Dataset $\mathcal{D}$, number of user types $K$, parameter $\alpha_{prior}$, learning rate $\beta$.
2: Initialize $\eta = (1/K, \ldots, 1/K)$, target network $\tilde{\theta} = \theta_0$
3: **for** $t$ in $\{0, \ldots, T\}$ **do**
4:     Sample a mini-batch $\mathcal{B} \sim \mathcal{D}$.
5:     **E-step.** Calculate posterior $\gamma_{i,k}$ for each example $(x_i, y_i) \in \mathcal{B}$ and user class $k \in [K]$:

$$\gamma_i(k) \leftarrow \frac{\eta_k \prod_{j=1}^M \sigma_{\tilde{\theta}}(x_{i,j}, y_{i,j}, k)}{\sum_{\ell=1}^K \eta_\ell \prod_{j=1}^M \sigma_{\tilde{\theta}}(x_{i,j}, y_{i,j}, \ell)}.$$

6:     **M-step.** Update parameters $\theta, \eta$:
$$\eta \leftarrow \alpha_{prior}\eta + (1 - \alpha_{prior})\mathbb{E}_{i \sim B}[\gamma_i]$$
$$\theta_{t+1} = \theta_t - \beta\nabla\mathcal{L}(\theta_t)$$

7:     Every $n$ steps: Update target network $\tilde{\theta} \leftarrow \theta_t$
8: **end for**
9: **Return:** model weights $\theta_T$.

---

## D.2. Losses and Models - Single Step

The EM framework is a general method applicable to various dataset and model types. In this work, we utilize two common single turn dataset types:

**Preference Model (HPS, Pick-a-Pic):** This model utilizes data samples consisting of two images generated by the same prompt, $x_i = (I_1^i, I_2^i, p^i)$, where the annotator indicates their preference, $y_i = (y_1^i \succeq y_2^i)$. To connect the model to the score function, we employ the Bradley-Terry (BT) model (Bradley & Terry, 1952), adapted from the RLHF reward learning framework to accommodate different user types. The user-specific BT model defines the probability that a user type $k$ prefers the first image over the other as:

$$\sigma_\theta^{pref}(x_i, y_i, k) = \frac{\exp\{s_\theta(I_1^i, p^i, k)\}}{\exp\{s_\theta(I_1^i, p^i, k)\} + \exp\{s_\theta(I_2^i, p^i, k)\}}. \tag{5}$$

In this case, the user loss function simplifies to:

$$\mathcal{L}^{pref}(\theta) = -\mathbb{E}_{i \sim \mathcal{B}, k \sim \gamma_i}\left[\sigma\big(s_\theta(I_1^i, p^i, k) - s_\theta(I_2^i, p^i, k)\big)\right]$$

where $\sigma$ represents the Sigmoid function.

**Relevance model (SAC):** This model provides a direct rating for an image-prompt pair, i.e., $x_i = (I^i, p^i)$, with the label being the integer rating level provided by the annotator, $y_i = s_i \in [S]$. We employ a simple user-conditional Gaussian model, where the score function acts as the mean estimator:

$$\sigma_\theta^{rel}(x_i, y_i = s_i, k) \propto \exp\left\{-\frac{(s_\theta(I^i, p^i, k) - s_i)^2}{2\nu^2}\right\}, \tag{6}$$

where $\nu$ is a predefined standard deviation. The user loss for the relevance model is:

$$\mathcal{L}^{rel}(\theta) = \frac{1}{2\nu^2}\mathbb{E}_{i \sim \mathcal{B}, k \sim \gamma_i}\left[(s_\theta(I^i, p^i, k) - s_i)^2\right].$$

When using relevance datasets, the rating levels are typically normalized to the range $[0, 1]$.

To maximize the number of training samples for the user model, a mixture of data types can be used. In the case of a batch from each type, $\mathcal{B} = (\mathcal{B}^{pref}, \mathcal{B}^{rel})$, a mixture loss function can be employed to balance between the two terms:

$$\mathcal{L}(\theta) = \frac{\kappa_1}{2\nu^2}\mathbb{E}_{i \sim \mathcal{B}^{rel}, k \sim \gamma_i}\left[(s_\theta(I^i, p^i, k) - s_i)^2\right] - \mathbb{E}_{i \sim \mathcal{B}^{pref}, k \sim \gamma_i}\left[\sigma(s_\theta(I_1^i, p^i, k) - s_\theta(I_2^i, p^i, k))\right],$$

where $\kappa_1$ controls the balance between the two terms.

## D.3. Losses and Models - Sequential

In this work, we introduce a novel multi-turn dataset. This dataset, generated by querying users across $H$ steps following the setting described in Section 2.2, consists of trajectories of user-agent interactions. Each sample $x_i = \{x_{i,t}\}_{t=1}^H$ has $x_{i,t} = \{p_0, \{I_{m,\ell}\}_{m=1,\ell=1}^{M,L}\}$ at each timestep $t$, with corresponding labels:

1. The chosen prompt index at each turn: $y_{i,t}^{cm} = c_{i,t} \in [L]$.

2. In-turn preferences between the best images for each prompt: $y_{i,t}^{in-pref} = \{(y_\ell^{i,t} \succeq y_{\ell'}^{i,t})\}_{\ell,\ell' \in \mathcal{Q}}$, where $\mathcal{Q}$ represents the set of all unordered pairs of size $|\mathcal{Q}| = \binom{L}{2}$.

3. Cross-turn preferences between chosen images at consecutive turns: $y_{i,t}^{cross-pref} = (y_\ell^{i,t} \succeq y_{\ell'}^{i,t-1})$, resulting in $H-1$ cross-turn preferences per sample.

Preferences within a sample are modeled by applying the BT model (Equation 5) to both in-turn and cross-turn preferences. User choices are modeled using our user choice model (Equation 4):

$$\sigma_\theta^{cm}(x_{i,t}, y_{i,t} = c_{i,t}, k) = \text{Softmax}(\tau_\theta \mathbf{s}_{i,t}^k)_{c_{i,t}}.$$

This leads to the following cross-entropy loss for the score function and choice model parameters:

$$\mathcal{L}^{cm}(\theta) = \mathbb{E}_{i,t \sim B, k \sim \gamma_i} \left[ -\log(\text{Softmax}(\tau_\theta \mathbf{s}_{i,t}^k)_{c_{i,t}}) \right].$$

In addition, we add a regularization term to stabilize the CLIP model finetuning by adding $\ell_2$ norm between the the original CLIP embeddings and the finetuned CLIP marked as $\mathcal{L}^{reg}(\theta)$.

The total multi-turn loss is then given by:

$$\mathcal{L}(\theta) = \mathcal{L}^{cm}(\theta) + \kappa_2 \mathcal{L}^{in-pref}(\theta) + \kappa_3 \mathcal{L}^{cross-pref}(\theta) + \kappa_4 \mathcal{L}^{reg}(\theta),$$

where $\kappa_2$, $\kappa_3$ and $\kappa_4$ are balancing coefficients.

## D.4. Architecture

Our score function utilizes pre-trained CLIP-ViT-L-14 (Radford et al., 2021) text and image encoders, augmented with user-specific encoders (one per user type). These user encoders transform CLIP embeddings into user-type-specific representations, capturing individual preferences for images and prompts. The final score for an image-prompt pair is calculated as the inner product of the user-specific image and text embeddings, scaled by a learned temperature parameter (see a schematic of our arch. in Figure 3).

We modified the CLIP encoders by removing the output l2-normalization. User preferences are modeled by a four-layer fully connected network with ReLU activation, structured as $d_{CLIP} - d_{CLIP} \times 2 - d_{CLIP} \times 4 - d_{CLIP} \times K$, where $d_{CLIP} = 768$ denotes the output dimension of the CLIP encoders. The final layer is partitioned into $K$ vectors, each of dimension $d_{CLIP}$, representing the different user types. These vectors act as residual offsets to the original CLIP output, individually added to it before a final $\ell_2$-normalization ensures each user score falls within the range $[0, 1]$. Each user score is multiplied by a learned temperature parameter. This architecture is illustrated in Figure 12. As for the temperature model, we use a two-layer fully connected network with ReLU activation, followed by a scaled Sigmoid function to bound the output in the range $[0, \tau_{max}]$. The network receives the utility score for each prompt candidate and produces the choice temperature scalar at the output.

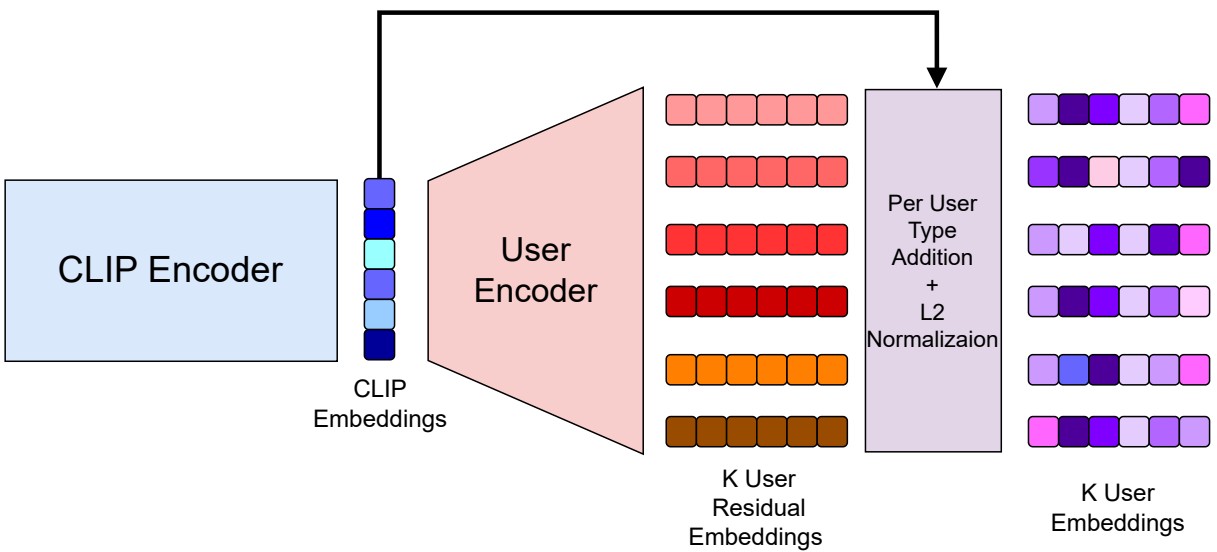

Figure 12: A diagram of our user encoder. The same framework is applied to both the text and image encoding process.

## D.5. Hyperparameters

Table 1: Main training hyperparameters

| | |
|---|---|
| Learning rate | Cosine annealing scheduler (lr=3e-4, T=10e3) (Loshchilov & Hutter, 2016) |
| Training steps | 50e3 |
| Batch size | 2048 |
| Update target network phase | 256 |
| Optimizer | AdamW (Loshchilov & Hutter, 2019) (weight decay = 1e-4) |
| $\kappa_1$ | 1 |
| $\alpha_{prior}$ | 0.999 |

Table 2: Fine-tuning hyperparameters

| | |
|---|---|
| Learning rate | Cosine annealing scheduler (lr=3e-7, T=10e3) (Loshchilov & Hutter, 2016) |
| Training steps | 50e3 |
| Batch size | 8 |
| Gradient norm clipping | 0.5 |
| Update target network phase | 256 |
| Optimizer | AdamW (Loshchilov & Hutter, 2019) (weight decay = 1e-2) |
| $\kappa_2$ | 0.01 |
| $\kappa_3$ | 1 |
| $\kappa_4$ | 0.1 |
| $\alpha_{prior}$ | 0.999 |
| $\tau_{\max}$ | 3 |

# E. PASTA Details

We employ PASTA, a novel framework designed for prompt selection in interactive image generation scenarios. PASTA utilizes Implicit Q-Learning (IQL) to train the candidate selector policy that selects optimal prompt expansions based on user interaction history. This section details the training process, including the IQL objective and adaptations for token-level training; the architecture and functionality of the candidate generator and selector components, which employ large language models (LLMs); and the hyperparameters used for training.

## E.1. Training

Generally, we would like to optimize the Q-Learning objective:

$$\mathcal{L}(\phi) = \mathbb{E}_{h_t, P_t, r_t, h_{t+1}, P_L^{t+1} \sim \mathcal{D}} \left[ \left( q_\phi(h_t, P_t) - r_t - \max_{P \in \mathcal{P}_L\left(P_L^{t+1}\right)} q_{\hat{\phi}}(h, P) \right)^2 \right],$$

where $\hat{\phi}$ is the target network and $P_L^t = \{p_i^t\}_{i=1}^{L_C}$ is the candidate set at time $t$. Decomposing the slate value function into an average of prompt value function we get

$$\mathcal{L}(\phi) = \mathbb{E}_{h_t, P_t, r_t, h_{t+1}, P_L^{t+1} \sim \mathcal{D}} \left[ \left( \frac{1}{L} \sum_{p \in P_t} f_\phi(h_t, p) - r_t - \max_{P \in \mathcal{P}_L\left(P_L^{t+1}\right)} q_{\hat{\phi}}(h, P) \right)^2 \right].$$

Although the value function is trained on the value of entire utterances, we enhance its robustness by also optimizing it on sub-prompts, focusing on tokens within the prompts that share the same target:

$$\mathcal{L}(\phi) = \mathbb{E}_{h_t, P_t, r_t, h_{t+1}, P_L^{t+1} \sim \mathcal{D}} \left[ \left( \frac{1}{L} \sum_{p \in P_t} \frac{1}{T_p} \sum_{t'=1}^{T_p} f_\phi(h_t, p^{1:t'}) - r_t - \max_{P \in \mathcal{P}_L\left(P_L^{t+1}\right)} q_{\hat{\phi}}(h, P) \right)^2 \right],$$

where $T_p$ is the (token) length of prompt $p$. Offline Reinforcement Learning struggles with overestimation of values for unseen state-action pairs when using the Bellman optimality operator for TD targets. *implicit Q-Learning (IQL)* (Kostrikov et al., 2021) addresses this by learning an $\alpha$-expectile value, $v_\psi$, which avoids the explicit maximization over the Q-function. The IQL objective is as follows:

$$\mathcal{L}(\phi, \psi) = \mathbb{E}_{h_t, P_t, r_t, h_{t+1} \sim \mathcal{D}} \left[ (q_\phi(h_t, P_t) - r_t - v_\psi(h_{t+1}))^2 + L_2^\alpha\left(q_{\hat{\phi}}(h_t, P_t) - v_\psi(h_t)\right) \right],$$

where $L_2^\alpha(x) = \left| \alpha - 1_{\{x<0\}} \right| x^2$, $\alpha \in [0.5, 1]$. Plugging in the decomposition of the value function and token-level training we get:

$$\mathcal{L}(\phi, \psi) = \mathbb{E}_{h_t, P_t, r_t, h_{t+1} \sim \mathcal{D}} \left[ \left( \left( \frac{1}{L} \sum_{p \in P_t} \frac{1}{T_p} \sum_{t'=1}^{T_p} f_\phi(h_t, p^{1:t'}) - r_t - v_\psi(h_{t+1}) \right)^2 + L_2^\alpha\left( \frac{1}{L} \sum_{p \in P_t} f_{\hat{\phi}}(h_t, p) - v_\psi(h_t) \right) \right) \right].$$

## E.2. Candidate Generator

We utilize a multimodal Large Language Model (LLM) based on Gemini 1.5 Flash [Team, 2024] as our candidate generator policy. This LLM takes the user's current interaction history as input and generates a set of $L_C$ prompt expansion candidates, categorized into distinct groups. To avoid overly long prompts, especially in later interaction steps, we impose a length constraint of $N_w^t$ words on the prompt at each step $t = 1, \ldots, H$. This constraint is dynamically adjusted according to the formula:

$$N_w^t = \frac{N_w^{t-1}(N_w^{max} - N_w^{t-1})}{H} t,$$

where $N_w^{max}$ represents the maximum permissible word count at the final step ($N_w^H \leq N_w^{max}$) and $N_w^0$ is the initial prompt's word count. Candidate generation employs nucleus sampling (Holtzman et al., 2019) with $p = 0.8$. The structure of the candidate generator prompts is illustrated in Figure 13.

---

**Candidate Generator Prompt Structure (example for t=3)**

You are an agent interacting with a user. The user wants to generate an image using an off-the-shelf diffusion model.
The user starts by providing a prompt for the diffusion model.
At every interaction step you (the agent) must decide on twenty-five (25) possible new prompts that would satisfy the user,
based on your understanding of their preferences given your interaction history with them.
Your prompts should be diverse enough to help you quickly capture the user preferences.
A new prompt could be a change of the prompt itself, or an expansion of the prompt.
Any new prompt should remain consistent with the original prompt.
At every interaction slate four (4) of the prompt candidates will be used to generate slates of four (4) images each,
for a total of sixteen (16) images presented to the user at every turn.
The user will then select exactly one of the four (4) slates of images (i.e., corresponding to one of the new prompts).
After five (5) interaction steps with the user, the user will choose one of the final image slates.
Your objective is to find an optimal slate of images for the user in the final interaction, based on their selections.

Output your slate in the following format. You should decide on five (5) categories, each listing exactly five (5) possible prompt candidates.
The format should look like so:
* <category_1>
- new_prompt_candidate_1
- new_prompt_candidate_2
...
- new_prompt_candidate_5
...
* <category_5>
- new_prompt_candidate_1
- new_prompt_candidate_2
...
- new_prompt_candidate_5

Below is the current interaction history with the user:
**Initial_prompt:** "{{ initial_prompt }}"
**Initial_image_output:** << image_samples >>
**AGENT:** Agent shows four prompts (with corresponding slates of images) to the user:
1. Prompt: "{{ prompt_expansion 1}}", Samples: << image samples >>
2. Prompt: "{{ prompt_expansion 2}}", Samples: << image samples >>
3. Prompt: "{{ prompt_expansion 3}}", Samples: << image samples >>
4. Prompt: "{{ prompt_expansion 4}}", Samples: << image samples >>
**USER**: User selects prompt number {{ image_number }} corresponding to the prompt: "{{ prompt_expansion }}"
**AGENT:** Agent shows four prompts (with corresponding slates of images) to the user:
1. Prompt: "{{ prompt_expansion 1}}", Samples: << image samples >>
2. Prompt: "{{ prompt_expansion 2}}", Samples: << image samples >>
3. Prompt: "{{ prompt_expansion 3}}", Samples: << image samples >>
4. Prompt: "{{ prompt_expansion 4}}", Samples: << image samples >>
**USER**: User selects prompt number {{ image_number }} corresponding to the prompt: "{{ prompt_expansion }}"

Based on the interaction history above, please provide twenty-five (25) prompt candidates to the final selected prompt.
Output should be divided into five (5) categories, each listing five (5) possible candidates, with the format described above.
The prompt candidates should be prompts that can be used as input in a diffusion model. The category titles won't be used as input.
One of the categories should be new prompts that are very different than the previous chosen prompt (for example rephrasing).
Any prompt candidate should be at most {{ num_words }} words long.

---

Figure 13: Template for the multimodal candidate generator prompt at step $t = 3$, with slate size $L = 4$, $L_C = 25$ candidates across five categories (colors for visual distinction).

### E.3. Candidate Selector

The candidate selector policy leverages the prompt-value function:

$$\pi_{S,\phi}(h) \in \underset{P \in \mathcal{P}_L\left(\{p_i\}_{i=1}^{L_C}\right)}{\arg\max} q_\phi(h, P) = \underset{P \in \mathcal{P}_L\left(\{p_i\}_{i=1}^{L_C}\right)}{\arg\max} \sum_{p \in P} f_\phi(h, p).$$

This policy is initialized with a pre-trained Gemma 2B model (Gemma-Team, 2024) for prompt-value estimation. In order to encourage diversity and exploration, we constraint the candidate selector to pick only up to one prompt from every category. Given an input $(h, p)$, represented as a token sequence of length $T^h + T^p$, the transformer produces an output of the same length, with vocabulary-sized logits for each token. We designate a specific logit, $\ell_q$, as the "q logit." The value for a history-prompt pair is then defined as the value of this q logit at the final position in the output sequence, corresponding to

the last input token:

$$f_\phi(h, p) = \{LLM([h, p]; \phi)\}_{\ell_q},$$

where $[\cdot, \cdot]$ denotes concatenation. We also employ the same model for the $\alpha$-expectile value estimator $v_\psi$ (setting $\psi \equiv \phi$):

$$v_\phi(h) = \{LLM(h; \phi)\}_{\ell_v},$$

where $\ell_v$ represents the $\alpha$-expectile value logit. Empirically, we observed optimal performance when $\ell_v$ and $\ell_q$ are identical. We prioritize efficiency and a small footprint for our prompt value function model, making Gemma 2B a suitable choice. Since Gemma 2B is a text-only model, we represent the history as text, encompassing the initial prompt, selected slates, and user-chosen prompts. When evaluating a history-prompt pair, the candidate prompt is also included in the input. The prompt template is detailed in Figure 14.

---

**History-Prompt Value Function Input Structure (example for t=3)**

<<INITIAL_PROMPT>>{{ initial prompt }}<<SELECTED_SLATE>>{{prompt_1}}||{{prompt_2}}||{{prompt_3}}||
{{prompt_4}}<<USER_CHOSEN_PROMPT>>{{ chosen_prompt }}<<SELECTED_SLATE>>{{prompt_1}}||{{prompt_2}}||{{prompt_3}}||
{{prompt_4}}<<USER_CHOSEN_PROMPT>>{{ chosen_prompt }}<<PROMPT_CANDIDATE>>{{ prompt_candidate }}

---

**History Value Function Input Structure (example for t=3)**

<<INITIAL_PROMPT>>{{ initial prompt }}<<SELECTED_SLATE>>{{prompt_1}}||{{prompt_2}}||{{prompt_3}}||
{{prompt_4}}<<USER_CHOSEN_PROMPT>>{{ chosen_prompt }}<<SELECTED_SLATE>>{{prompt_1}}||{{prompt_2}}||{{prompt_3}}||
{{prompt_4}}<<USER_CHOSEN_PROMPT>>{{ chosen_prompt }}

---

Figure 14: Text-only input templates for the value functions at step $t = 3$ with a slate size of $L = 4$. The top template corresponds to the history-prompt value function, while the bottom template corresponds to the prompt value function.

### E.4. Hyperparameters

Table 3: PASTA hyperparameters

| | |
|---|---|
| Learning rate | 1e-5 |
| Training steps | 1e4 |
| Batch size | 128 |
| Optimizer | Adafactor (Shazeer & Stern, 2018) (weight decay = 1e-2) |
| Gradient norm clipping | 1 |
| Update target network phase | 256 |
| Expectile parameter $\alpha$ | 0.7 |
| $\ell_q$ | 651 |
| $\ell_v$ | 651 |
| $L$ | 4 |
| $M$ | 4 |
| $L_C$ | 25 |
| Number of categories | 5 |
| $H$ | 5 |
| $N_w^{max}$ | 62 |

# F. Additional Experiments

This section presents additional experiments conducted to further validate the effectiveness of our proposed framework. We first evaluate the performance of PASTA with simulated users under different training data regimes and reward settings. Subsequently, we showcase PASTA's ability to adapt to diverse user preferences by visualizing interactions with various user types on abstract image generation prompts.

## F.1. Simulated Users Experiment

We performed an experiment with simulated users, leveraging our trained user model. Specifically, we evaluated our trained value models across three configurations: (1) trained using only human rater data, (2) trained using only simulated data, and (3) trained with a mixture of both. These models were compared to a random sampling policy to isolate the effect of the trained value models. All approaches employed the same candidate generator, which is based on the multimodal Gemini 1.5 Flash (Gemini-Team, 2024). The experiment was conducted in two reward settings: (1) **dense reward**, where the utility function provided rewards at every step, and (2) **sparse reward**, where rewards were given only at the final step, focusing on scenarios where only the final outcome matters. We evaluated all methods over 1000 trajectories, each generated as described in Section 2.2. The results are summarized in the table below:

|        | Random           | Human            | Simulated            | Mixed             |
|--------|------------------|------------------|----------------------|-------------------|
| Sparse | $7.38 \pm 1.13$  | $8.7 \pm 0.11$   | $\mathbf{9.26 \pm 0.07}$   | $8.81 \pm 0.06$   |
| Dense  | $38.57 \pm 5.81$ | $43.49 \pm 0.54$ | $\mathbf{46.35 \pm 0.38}$  | $44.03 \pm 0.32$  |

The results demonstrate that all value-based methods outperform the random sampling policy. Notably, the value model trained exclusively on simulated data achieves the best performance. This outcome aligns with expectations, as the simulated data matches the dynamics of the simulated user model precisely. While human data offers some improvement in isolation, its divergence from the simulated dynamics reduces the effectiveness when combined with simulated training data.

## F.2. Abstract Prompts with Simulated Users

To visualize the differences in user preferences and PASTA's ability to adapt to them, we ran PASTA with various user types using broad, abstract prompts such as "an image of happiness." These prompts imposed minimal constraints on the user model, allowing each user type to express its unique preferences freely. For some user types, we observed a distinct preference emerging, favoring specific styles or content. All users starts with the same prompt and initial images. We present only the images and their corresponding prompt-expansion at the last 5-th step. Each color represent a different user type. The results are as follow:

**"An image of happiness":**

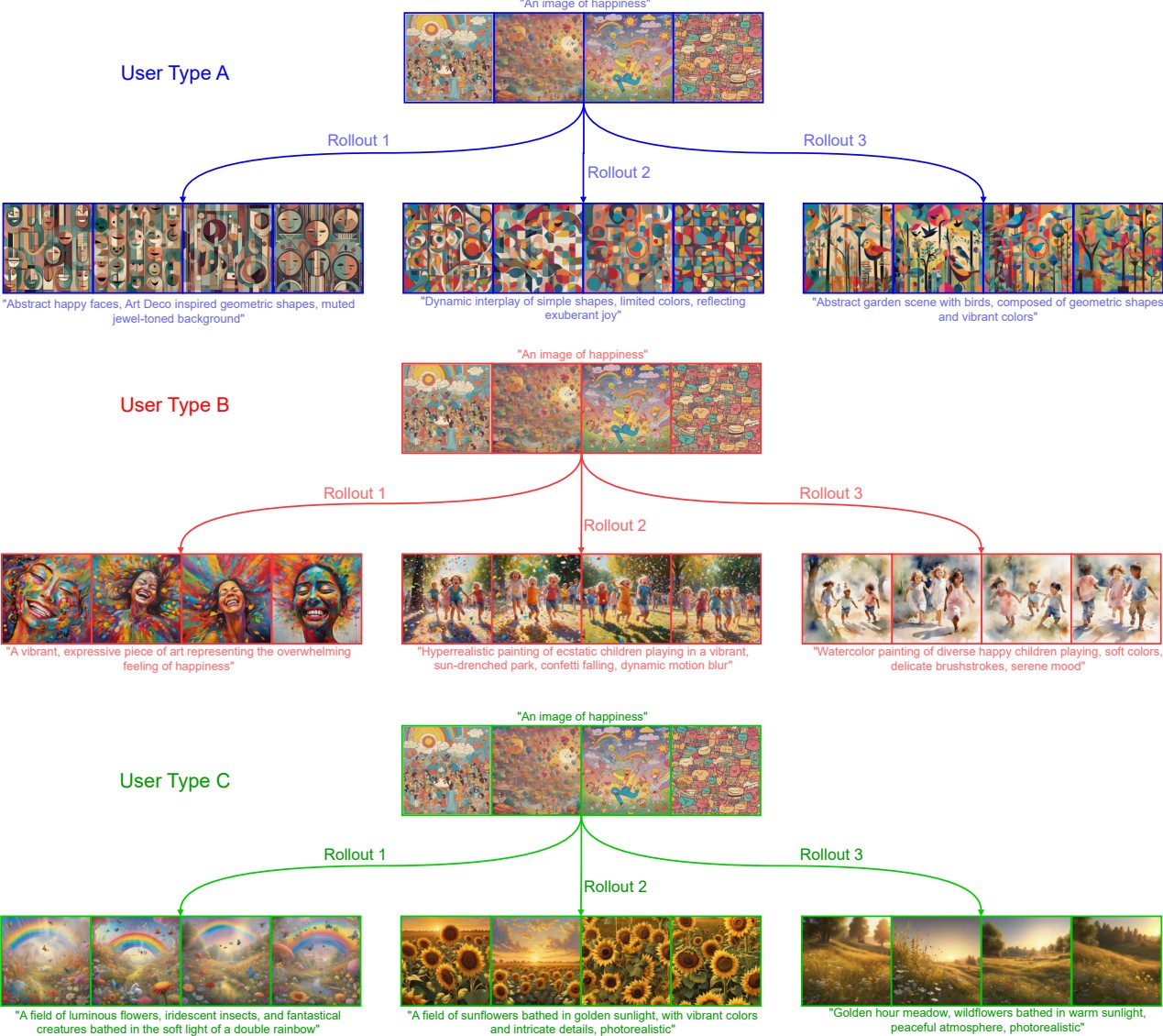

**"An image of love":**

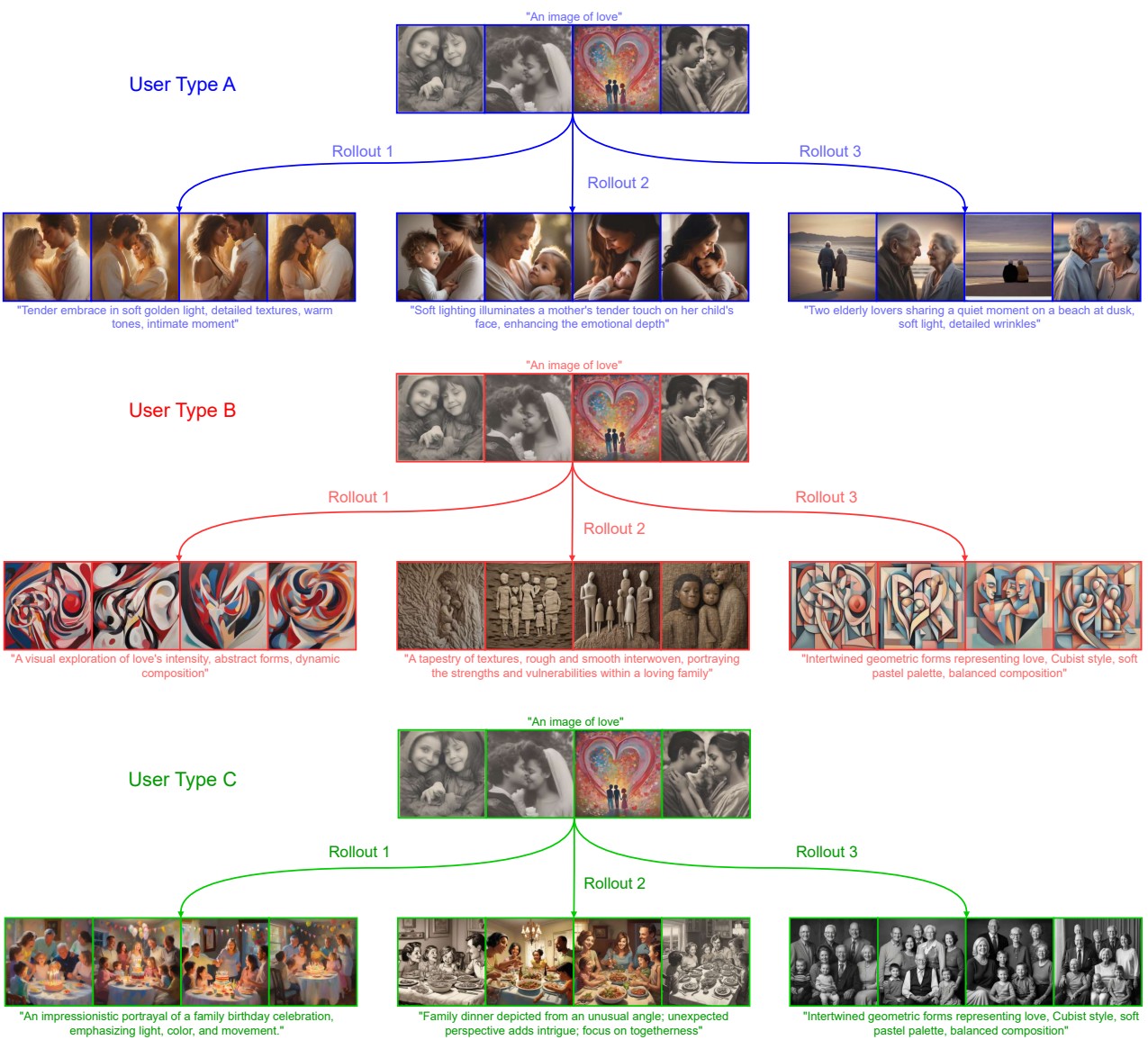

**"An image of hope":**

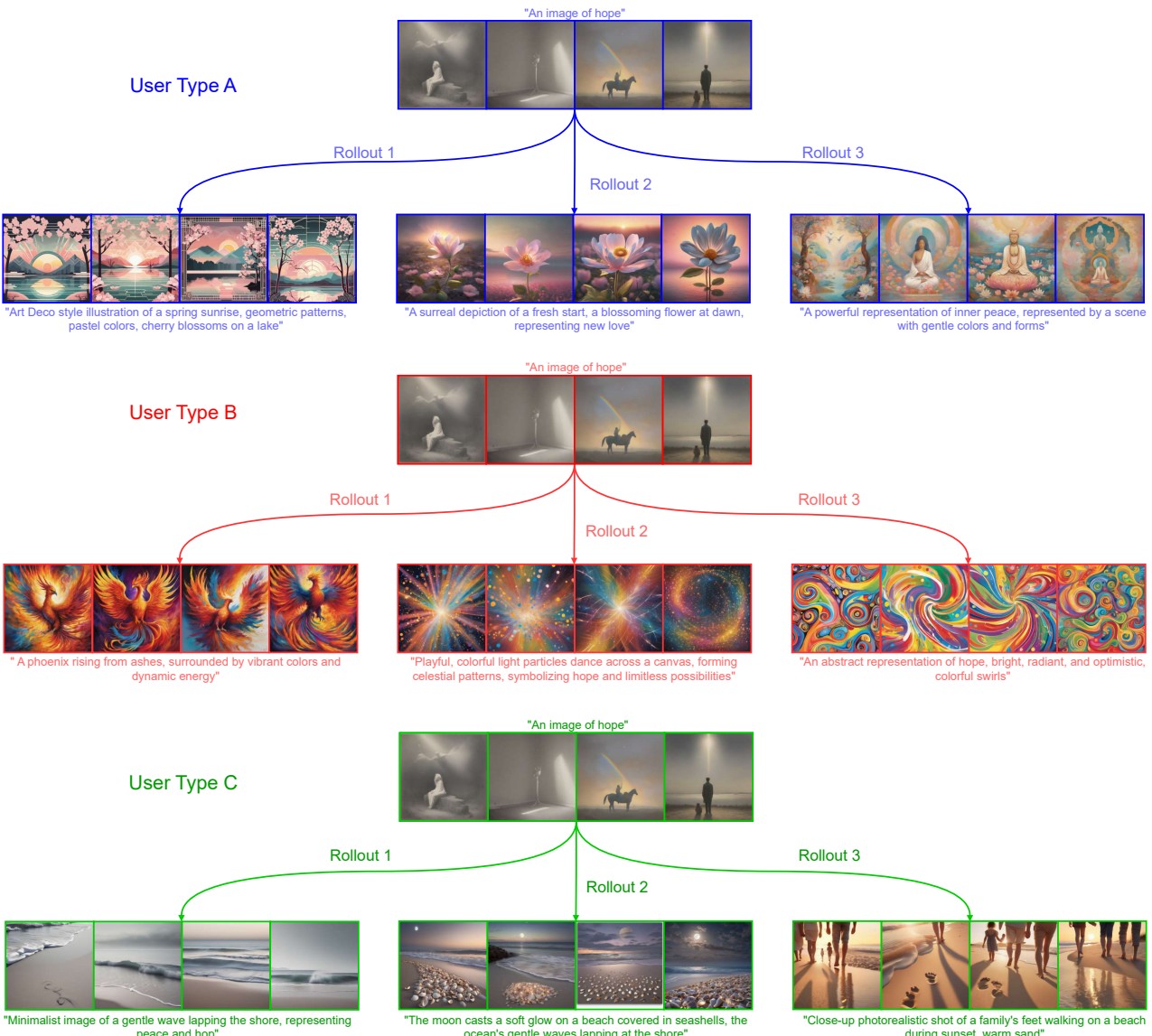

# G. PASTA Examples

We illustrate a complete 5-turn interaction between our user model and the PASTA agent with examples. Each example begins with the initial prompt and images. Each turn shows the agent's 25 generated candidate prompts (organized into 5 categories) on the left. Four of these prompts (highlighted in red) are offered to the user, selected by a sampling policy based on a value function. The four generated images corresponding to each proposed prompt are displayed below. The prompt ultimately chosen by the user is highlighted in blue and shown above the image set.

**Example #1:**



**Initial prompt:**
"A train is going down the train tracks"



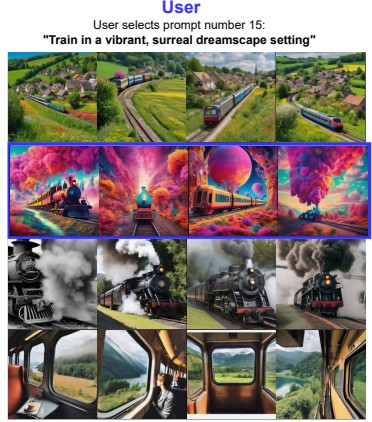

•••••••••••••••••••••••••••••••••••••••••••••••••••••••• **Turn 1** ••••••••••••••••••••••••••••••••••••••••••••••••••••••••

**Agent**

**Category: Variations in Lighting and Time of Day:**
1. A train at sunset, on tracks.
2. A train in the morning mist.
3. A train under a stormy sky.
4. A train at night, glowing lights.
5. A train in bright sunlight.
**Category: Variations in Train Type and Composition:**
6. A long freight train on tracks.
7. A sleek modern passenger train.
8. An old steam train on tracks.
9. A colorful toy train on tracks.
10. A double-decker train on tracks.
**Category: Variations in Environment and Setting:**
11. Train on a mountain track.
12. Train in a desert landscape.
13. Train crossing a bridge.
14. Train passing through a forest.
15. Train in a snowy landscape.
**Category: Completely Different Prompts (Rephrasing):**
16. Railroad journey, scenic view.
17. Metal beast on steel rails.
18. Locomotive speeding onward.
19. Tracks vanishing in distance.
20. Steel wheels on iron rails.
**Category: Adding Artistic Styles and Effects:**
21. Train on tracks, impressionist style.
22. Train, hyperrealistic rendering.
23. Train on tracks, painted by Monet.
24. Train, detailed pencil sketch.
25. Train on tracks, vibrant colors.

**User**
User selects prompt number 16:
**"Railroad journey, scenic view."**

•••••••••••••••••••••••••••••••••••••••••••••••••••••••• **Turn 2** ••••••••••••••••••••••••••••••••••••••••••••••••••••••••

**Agent**

**Category: Expanding the Scene:**
1. Railroad tracks winding through a lush green valley, sunlit.
2. Train disappearing into a mountain tunnel, dramatic lighting.
3. A vintage steam train crossing a high bridge over a river.
4. Railroad tracks stretching to the horizon, sunset over fields.
5. Train passing a quaint countryside village, wildflowers.
**Category: Focusing on the Train:**
6. Majestic steam locomotive pulling luxurious passenger cars.
7. Modern high-speed train speeding through a vibrant landscape.
8. Freight train carrying colorful containers, scenic backdrop.
9. Close-up of a train's wheels turning on the tracks, detailed.
10. A weathered, vintage train car in a beautiful setting.
**Category: Artistic Styles:**
11. Railroad landscape, painted in the style of Van Gogh.
12. Impressionist painting of a train journey through fields.
13. Photorealistic train passing through a stunning canyon.
14. A whimsical, cartoon-style train in a fantasy landscape.
15. Train in a vibrant, surreal dreamscape setting.
**Category: Different Perspectives:**
16. Aerial view of a train snaking through a winding valley.
17. View from inside a train car, looking out at the scenery.
18. A long exposure shot of a train moving at night, light trails.
19. Close-up of the train's engine, steam billowing dramatically.
20. View from the train's cab, looking down the tracks.
**Category: Rephrasing and Variations:**
21. Scenic railroad journey through diverse landscapes.
22. Picturesque train ride, breathtaking views from the window.
23. Railroad adventure, panoramic scenery.
24. Journey by rail, spectacular countryside.
25. Train on rails, beautiful and idyllic nature.

**User**
User selects prompt number 15:
**"Train in a vibrant, surreal dreamscape setting"**

················································ **Turn 3** ····················································

**Agent**

**User**

**Category: Surreal Landscapes & Trains:**
1. A steam train chugging through a field of giant, iridescent flowers under a swirling galaxy.
2. A whimsical train journey through a candy-colored forest, lollipop trees, and marshmallow clouds.
3. A futuristic train gliding over a rainbow bridge in a city made of crystals and floating islands.
4. An antique train winding through a valley of vibrant, bioluminescent plants at twilight.
**5. A surreal train station nestled within a giant, glowing mushroom forest.**
**Category: Emphasis on Vibrancy & Color:**
6. A vibrant, hyperrealistic train speeding through a landscape of impossibly bright colors.
7. A psychedelic train ride through a kaleidoscope of swirling patterns and neon lights.
8. A train made of stained glass, traversing a landscape of equally colorful stained glass.
9. A cartoonish train with exaggerated features, traveling through a bright, playful world.
10. A majestic train, with golden carriages, passing through a field of glowing sunflowers.
**Category: Focus on Dreamlike Atmosphere:**
11. A lone train disappearing into a hazy, dreamlike fog, leaving a trail of shimmering stardust.
**12. A train floating effortlessly above a sea of clouds, bathed in soft, ethereal light.**
13. A vintage train gliding silently through a landscape painted in watercolor washes.
14. A dreamlike train station, where clocks melt and platforms float, bathed in moonlight.
15. A train composed of shifting, morphing shapes, moving through a constantly changing landscape.
**Category: Different Interpretations of the Prompt:**
16. A train journey through an impossible landscape, defying gravity and perspective.
17. The feeling of riding a train in a vibrant dream, expressing freedom and wonder.
18. A dream sequence, a train traveling through a bizarre, otherworldly environment.
19. A surreal, dreamlike depiction of a train journey, full of vibrant colors and unusual imagery.
**20. A train's passage through a vibrant dreamscape, emphasizing the contrast between reality and fantasy.**
**Category: Adding Specific Artistic Styles:**
21. A train in a vibrant surreal dreamscape, in the style of a 1950s cartoon.
**22. A vibrant surreal dreamscape with a train, painted in the style of Salvador Dali.**
23. A vibrant surreal dreamscape featuring a train, rendered in the style of Alphonse Mucha.
24. A train traveling through a vibrant surreal dreamscape, with a focus on Art Nouveau style.
25. A vibrant surreal dreamscape with a train, using bold, expressive brushstrokes in an Impressionistic style.

User selects prompt number 5:
**"A surreal train station nestled within a giant, glowing mushroom forest"**

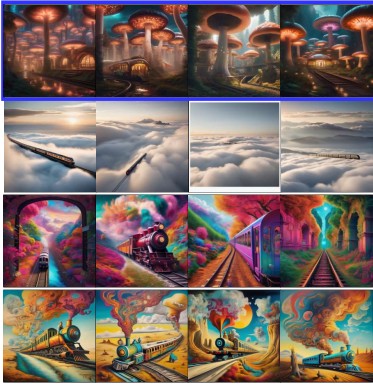

················································ **Turn 4** ····················································

**Agent**

**User**

**Category: Vibrant Mushroom Forests & Trains:**
1. A whimsical train station nestled amongst bioluminescent mushrooms, with a steampunk aesthetic.
2. A vibrant, colorful train speeding through a fantastical forest of giant glowing mushrooms.
3. A surreal train journey through a mushroom forest, with playful, cartoonish mushrooms.
**4. A detailed illustration of a train emerging from a glowing mushroom cavern, magical atmosphere.**
5. A train puffing steam, passing through a dense forest of giant, otherworldly mushrooms.
**Category: Surreal Landscapes & Trains:**
6. A train gliding across a rainbow bridge in a surreal, dreamlike landscape.
7. A train traversing a landscape of floating islands, vibrant colors, and impossible architecture.
8. A surreal train journey through a landscape painted in the style of Van Gogh, vibrant colors.
9. A hyperrealistic train in a fantastical landscape, filled with impossible flora and fauna.
10. A train disappearing into a swirling vortex of colors in a surreal, dreamlike landscape.
**Category: Dali-esque Trainscapes:**
11. A melting clock train in a Dali-esque landscape, with distorted perspective and melting forms.
12. A train with elongated limbs and surreal features in a Dali-inspired dreamscape.
**13. A train composed of melting clocks and surreal objects in a Dali-esque landscape.**
14. A train traversing a desert landscape with melting rocks and distorted space in Dali's style.
15. A surreal train scene with distorted perspective and soft, flowing lines in the style of Dali.
**Category: Ethereal & Dreamlike Trains:**
**16. A train floating through a starry night sky, with ethereal lights and glowing constellations.**
17. A lone train moving through a field of luminous flowers in a soft, ethereal light.
18. A dreamy train scene with pastel colors, soft clouds, and a peaceful, tranquil atmosphere.
19. A train gently gliding through a soft, misty landscape, with a serene and peaceful atmosphere.
20. A train traveling through a sea of clouds, bathed in soft, golden light, peaceful and serene.
**Category: Completely Different Prompts (Rephrasing):**
21. A detailed photograph of a modern high-speed train passing through a rural area.
**22. A realistic depiction of a vintage steam train chugging along a mountain pass.**
23. A classic black and white photo of a train passing through a bustling city.
24. An image of a freight train transporting goods along a long railway line.
25. A simple drawing of a train on tracks in a straightforward, realistic style.

User selects prompt number 13:
**"A train composed of melting clocks and surreal objects in a Dali-esque landscape"**

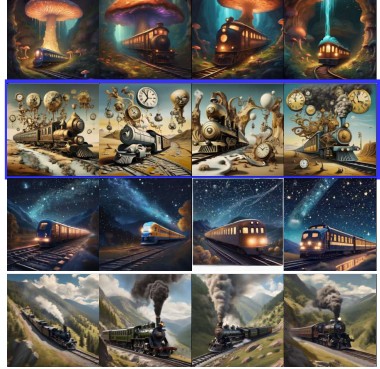

················································ **Turn 5** ····················································

**Agent**

**User**

**Category: Dali-esque Variations:**
**1. A surreal train speeding through a landscape of melting clocks and distorted figures, in the style of Salvador Dali.**
2. A train composed entirely of melting clocks, surreal objects, and impossible architecture, a Dali-esque dreamscape.
3. A Dali-esque train journey through a desert landscape, with elongated shadows and distorted perspectives.
4. A train with elongated limbs and melting forms, traversing a bizarre, dreamlike landscape, in the style of Salvador Dali.
5. A surreal train station, with distorted architecture and melting clocks, in the style of Salvador Dali, hyperrealistic.
**Category: Enhanced Surrealism:**
**6. A train made of liquid metal, flowing through a landscape of shifting colors and impossible geometries.**
7. A train gliding through a vibrant, surreal dreamscape filled with fantastical creatures and impossible structures.
8. A train passing through a swirling vortex of colors and shapes, a surreal and psychedelic journey.
9. A train emerging from a giant, melting clock, surrounded by distorted figures and impossible objects.
10. A train journey through a hyperrealistic, surreal landscape, with unexpected elements and juxtapositions.
**Category: Focus on the Train:**
11. Detailed close-up of a surreal train engine, with melting clocks and intricate details.
12. A surreal train's interior, filled with melting furniture, distorted mirrors, and bizarre passengers.
13. The perspective of someone sitting inside a Dali-esque train, witnessing the surreal landscape.
14. A highly detailed rendering of a surreal train, emphasizing textures and materials.
15. A surreal train, focusing on its unusual design and mechanics.
**Category: Different Artistic Styles:**
16. A train in a surreal landscape, painted in the style of René Magritte, with a focus on unexpected juxtapositions.
17. A train journey through a surreal landscape, rendered in the style of a woodcut print, with bold lines and textures.
18. A surreal train scene in the style of Max Ernst, featuring collage-like elements and unexpected combinations.
19. A surreal train depicted in a vibrant, pop art style, with bold colors and graphic elements.
**20. A surreal train in a cubist landscape, emphasizing fragmented forms and multiple perspectives.**
**Category: Rephrasing & Simplification:**
21. A bizarre, dreamlike train in a surreal landscape.
**22. A surreal train, simple prompt.**
23. A melting clock train in a Dali-esque world.
24. Surreal train in a distorted landscape.
25. A dreamlike train journey through a strange land.

User selects prompt number 22:
**"An ethereal underwater scene depicting a queen, her crown formed by bioluminescent sea creatures, radiating an enchanting glow."**

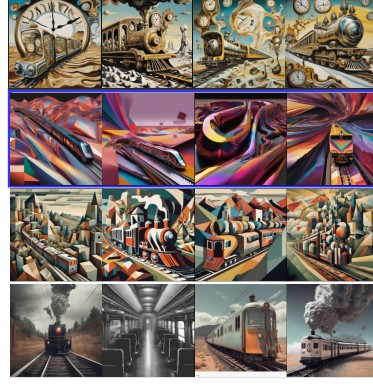

## Example #2:

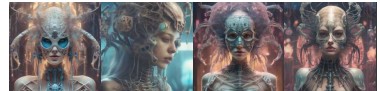



**Initial prompt:**

"A princess wearing an animal skull mask with a ribcage skeleton and
cybernetic jellyfish features"


••••••••••••••••••••••••••••••••••••••••••• Turn 1 •••••••••••••••••••••••••••••••••••••••••••

**Agent**

**Category: Enhanced Detail & Lighting:**
1. Close-up portrait, princess, animal skull mask, cybernetic jellyfish, intricate details, dramatic lighting.
2. Princess, skull mask, jellyfish cybernetics, ethereal glow, hyperrealistic, close-up.
3. Macro portrait, princess, bone mask, bioluminescent jellyfish, sharp focus, rich colors.
4. Detailed portrait, princess, ornate skull mask, glowing cybernetic jellyfish, dark background.
**5. Close-up, princess, animal skull headdress, bio-mechanical jellyfish, cinematic lighting.**
**Category: Alternative Mask & Cybernetics:**
6. Princess portrait, elegant bird skull mask, subtle cybernetic enhancements, soft lighting.
7. Close-up, princess, futuristic insect mask, integrated cybernetic jellyfish, vibrant colors.
8. Macro portrait, princess, ornate dragon skull mask, glowing cybernetic implants, blurred background.
**9. Princess, skeletal mask, sleek cybernetic arms, ethereal background, detailed texture.**
10. Portrait, princess, geometric skull mask, minimalist cybernetic features, sharp focus.
**Category: Variations in Background & Setting:**
**11. Princess, skull mask, cybernetic jellyfish, underwater scene, shimmering bubbles.**
12. Portrait, princess, bone mask, cybernetic enhancements, mystical forest background.
13. Close-up, princess, animal skull mask, jellyfish cybernetics, futuristic cityscape backdrop.
14. Princess, skull mask, bioluminescent jellyfish, otherworldly nebula background.
15. Macro portrait, princess, ornate mask, cybernetic elements, enchanted garden setting.
**Category: Different Artistic Styles:**
16. Princess, skull mask, cybernetic jellyfish, art nouveau style, elegant and flowing lines.
17. Portrait, princess, bone mask, cybernetic enhancements, dark fantasy art style.
18. Close-up, princess, animal skull mask, jellyfish cybernetics, photorealistic style.
19. Princess, skull mask, bioluminescent jellyfish, gothic art style, dramatic lighting.
20. Macro portrait, princess, ornate mask, cybernetic elements, digital painting style.
**Category: Completely Different Concept (Rephrasing):**
21. A regal woman, adorned with intricate bone jewelry and glowing bioluminescent accents.
**22. A mysterious figure, partially obscured by a fantastical mask, with subtle technological elements.**
23. A powerful queen, with cybernetic enhancements blending seamlessly with organic bone structures.
24. A captivating portrait of a woman, combining elegance with futuristic bio-mechanical elements.
25. An ethereal being, blending organic and synthetic elements in a harmonious composition.

**User**

User selects prompt number 5:
**"Close-up, princess, animal skull headdress, bio-
mechanical jellyfish, cinematic lighting"**

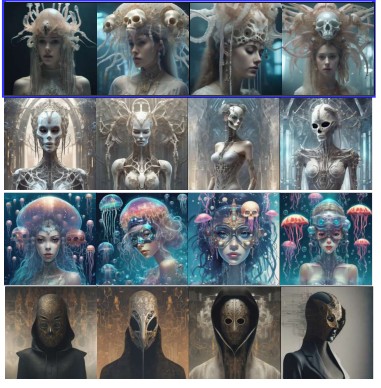

••••••••••••••••••••••••••••••••••••••••••• Turn 2 •••••••••••••••••••••••••••••••••••••••••••

**Agent**

**Category: Refined Details & Lighting:**
1. Close-up portrait: princess, intricate skull headdress, bioluminescent jellyfish, dramatic lighting, 8k resolution.
2. Princess in ornate skull mask, bio-mechanical jellyfish crown, deep shadows, sharp focus, cinematic lighting.
3. Close-up, princess, detailed animal skull mask, glowing jellyfish, volumetric lighting, dark background.
**4. Macro portrait: princess, skull headdress with jellyfish, chiaroscuro lighting, ethereal atmosphere, hyperrealistic.**
5. Close-up, princess, skull mask, bio-mechanical jellyfish, cinematic lighting, rich textures.
**Category: Variations on the Jellyfish:**
6. Princess, ornate skull mask, bioluminescent jellyfish tendrils, dark background, cinematic lighting.
7. Close-up portrait, princess, skull headdress, iridescent jellyfish, dark moody ambiance, sharp details.
8. Princess, skeletal mask, cybernetic jellyfish swarm, deep ocean setting, cinematic lighting.
**9. Close-up: princess, skull headdress, bio-mechanical jellyfish, neon lighting, underwater effect.**
10. Princess, skull mask, pulsating jellyfish, dark background, dramatic lighting, 8k resolution.
**Category: Emphasis on the Skull:**
11. Close-up portrait, princess, intricately carved skull mask, jellyfish accents, cinematic lighting.
12. Princess, detailed animal skull mask, bio-mechanical jellyfish, dark fantasy art, dramatic lighting.
13. Close-up, princess, ornate skull headdress, bio-mechanical jellyfish, dark fantasy art, sharp focus.
14. Princess, realistic skull mask, cybernetic jellyfish, dark background, cinematic lighting, detailed texture.
**15. Macro shot: princess, decaying skull mask, bio-mechanical jellyfish, dramatic lighting, ethereal feel.**
**Category: Completely Different Style:**
16. Elegant princess, futuristic crown, subtle technological details, soft lighting, pastel colors.
17. Portrait of a regal woman, ethereal crown, flowing gown, soft light, serene expression.
18. A powerful queen, majestic pose, ornate jewelry, warm lighting, rich colors.
19. Young woman, delicate flower crown, flowing dress, soft light, gentle expression.
20. Beautiful woman, simple headband, natural light, soft focus, dreamy atmosphere.
**Category: Exploring Backgrounds & Settings:**
**21. Close-up, princess, skull headdress, bio-mechanical jellyfish, foggy forest background.**
22. Princess, skull mask, jellyfish crown, cinematic lighting, ancient ruins background.
23. Close-up portrait, princess, skull headdress, bio-mechanical jellyfish, underwater scene.
24. Princess, skull mask, jellyfish, cinematic lighting, gothic architecture background.
25. Close-up: princess, skull headdress, bio-mechanical jellyfish, starry night sky background.

**User**

User selects prompt number 9:
**"Close-up: princess, skull headdress, bio-mechanical
jellyfish, neon lighting, underwater effect"**

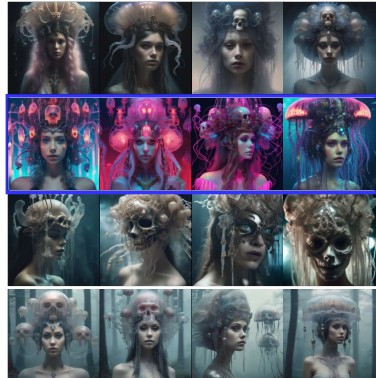

**Agent**

**Category: Neon Dreamscapes:**
1. Princess, skull jellyfish crown, neon coral reef, vibrant underwater city.
**2. Underwater princess, glowing jellyfish headdress, neon lights, art deco style.**
3. Cyberpunk princess, skull mask with neon jellyfish, futuristic underwater city.
4. Princess in neon-lit underwater cave, skull headdress, bioluminescent jellyfish.
5. Electric princess, skull and jellyfish crown, neon ocean, 80s synthwave aesthetic.
**Category: Subtle Shifts in Lighting and Background:**
6. Close-up portrait: princess, skull headdress, bioluminescent jellyfish, soft neon glow, dark background.
7. Princess, skull jellyfish crown, underwater scene, subtle neon accents, calm atmosphere.
8. Underwater princess, skull headdress, bio-mechanical jellyfish, pastel neon, serene background.
9. Princess, skull and jellyfish crown, deep ocean, cool neon hues, mysterious ambiance.
10. Close-up, princess, skull headdress, jellyfish, dim neon lighting, smoky underwater cave.
**Category: Variations on the Jellyfish:**
**11. Princess, skull headdress, glowing jellyfish tentacles, neon lights, dark ocean.**
12. Princess, skull crown, iridescent jellyfish, neon underwater landscape, detailed textures.
13. Underwater princess, skull headdress, bioluminescent jellyfish swarm, neon glow.
14. Close-up: princess, skull mask, intricate jellyfish headdress, neon coral, deep ocean.
15. Princess, skull headdress, bio-mechanical jellyfish, neon kelp forest, flowing fabric.
**Category: Completely Different Style:**
16. Portrait of a regal woman, ornate mask, intricate jewelry, rich textures, dramatic lighting.
**17. Elegant woman, mysterious headdress, flowing gown, detailed portrait, studio lighting.**
18. A woman with a captivating gaze, elaborate headdress, rich colors, painterly style.
19. Close-up portrait, strong female figure, enigmatic expression, rich textures, studio setting.
20. Fashion portrait, stylish woman, unique headpiece, luxurious fabrics, sophisticated look.
**Category: Emphasizing Texture and Detail:**
21. Hyperrealistic portrait: princess, detailed skull headdress, bio-mechanical jellyfish, neon lighting, underwater scene.
22. Close-up, princess, skull headdress, intricate jellyfish details, neon lights, underwater world.
**23. Princess, skull jellyfish crown, highly detailed textures, neon lighting, underwater cave.**
24. Macro portrait: princess, textured skull mask, bio-mechanical jellyfish, neon glow, underwater rocks.
25. Detailed portrait: princess, skull headdress, realistic jellyfish, neon accents, ocean floor.

**User**
User selects prompt number 23:
**"Princess, skull jellyfish crown, highly detailed textures, neon lighting, underwater cave"**

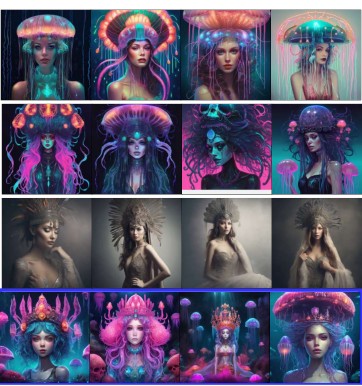

**Agent**

**Category: Neon Underwater Princesses:**
**1. A princess adorned with a bioluminescent jellyfish crown, swimming in a vibrant neon coral reef, intricate details, 8k resolution.**
2. Princess in an underwater palace, wearing a jellyfish headdress that glows with neon light, surrounded by glowing flora and fauna.
3. Portrait of a princess, her face partially obscured by a neon jellyfish crown, underwater scene, hyperrealistic details.
4. An ethereal princess with flowing hair, a jellyfish headdress emitting neon light, swimming gracefully amidst glowing jellyfish.
5. Close-up of a princess wearing a skull-shaped jellyfish crown, vibrant neon lighting, underwater cave setting, detailed textures.
**Category: Dark Fantasy Princesses:**
6. A princess in a dark underwater grotto, wearing a crown of glowing jellyfish and skeletal elements, mysterious and ethereal.
7. Dark fantasy portrait: a princess with a jellyfish crown made of bone and neon lights, surrounded by deep sea creatures.
8. A gothic princess, wearing a headdress crafted from decaying jellyfish and skulls, bathed in neon underwater light, dramatic pose.
9. A mysterious underwater princess, her face veiled by a jellyfish headdress with glowing neon accents, dark and moody atmosphere.
10. Portrait of a princess in a dark, underwater cavern, wearing a crown of bioluminescent jellyfish, surrounded by shadows and glowing accents.
**Category: Detailed Jellyfish Headwear:**
11. Extremely detailed portrait of a princess wearing an intricate jellyfish headdress, neon lighting, underwater setting, realistic textures.
**12. Close-up, princess with a jellyfish crown, featuring highly detailed tentacles and bioluminescent patterns, underwater scene, 8k resolution.**
13. Macro shot: princess, ornate jellyfish headdress with intricate skeletal details, neon lighting, underwater environment, sharp focus.
14. Princess portrait: elaborate jellyfish crown with glowing neon accents, detailed textures, underwater background, cinematic lighting.
15. Highly detailed image of a princess wearing a jellyfish crown, showcasing the intricate textures and neon glow, underwater environment.
**Category: Alternative Interpretations:**
**16. A regal figure, partially obscured by a mask of intricately woven light and shadow, with bioluminescent accents, underwater, 8k resolution.**
17. A captivating woman, her face partially hidden by a fantastical headpiece reminiscent of deep-sea life, bathed in ethereal light.
18. A mysterious and elegant woman with a headpiece combining natural and technological elements, emitting a soft, otherworldly glow.
19. An enigmatic figure in an underwater setting, adorned with a crown of ethereal light and intricate, bioluminescent patterns.
20. A vision of elegance and mystery: a woman with a unique headpiece inspired by marine life, emitting a gentle, glowing light.
**Category: Stylistic Variations:**
21. Art Deco style portrait: a princess wearing a geometric jellyfish crown, vibrant neon colors, underwater setting, elegant composition.
22. Surrealist portrait: a princess with a melting jellyfish crown, neon lighting, dreamlike underwater scene, unexpected elements.
23. Impressionistic underwater portrait: a princess with a jellyfish crown, soft focus, neon colors, flowing lines and shapes.
**24. A princess with a jellyfish headdress in the style of Alphonse Mucha, elegant lines, soft neon lighting, underwater background.**
25. A princess wearing a jellyfish crown, painted in the style of Gustav Klimt, with gold and vibrant neon accents, underwater scene.

**User**
User selects prompt number 12:
**"Close-up, princess with a jellyfish crown, featuring highly detailed tentacles and bioluminescent patterns, underwater scene, 8k resolution"**

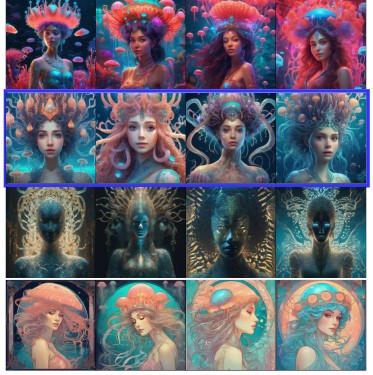

**Agent**

**Category: Refined Details & Lighting:**
1. Close-up portrait of a princess wearing a bioluminescent jellyfish crown, intricate details on tentacles, soft underwater lighting, 8k resolution, hyperrealistic.
2. Princess with a jellyfish crown, detailed tentacles with iridescent scales, soft blue and pink lighting, underwater scene, serene expression.
3. Underwater portrait: princess, jellyfish crown with glowing accents, subtle shimmering particles, cinematic lighting, highly detailed textures.
4. Close-up, princess, jellyfish crown, vibrant coral reef background, realistic rendering, sharp focus, detailed textures, soft lighting.
**5. Princess, jellyfish crown, realistic jellyfish anatomy, submerged in crystal clear water, natural lighting, ethereal atmosphere.**
**Category: Stylistic Variations:**
6. Art Nouveau princess with a jellyfish crown, flowing lines, pastel colors, underwater setting, elegant composition.
**7. Princess, jellyfish crown, Art Deco style, geometric patterns, neon lighting, underwater Art Deco architecture.**
8. Surrealist portrait: princess, melting jellyfish crown, dreamlike underwater environment, vibrant colors, distorted perspective.
9. Impressionist princess, jellyfish crown, soft brushstrokes, underwater scene, blurred background, pastel hues.
10. Princess, jellyfish crown, in the style of Gustav Klimt, gold leaf accents, underwater scene, shimmering effect.
**Category: Environmental Changes:**
11. Princess, jellyfish crown, deep ocean trench, bioluminescent creatures, dark mysterious background, 8k resolution.
12. Princess, jellyfish crown, glowing coral reef, diverse marine life, vibrant colors, shallow water, sunlight filtering through.
13. Princess, jellyfish crown, ancient sunken city, ruins, mystical underwater atmosphere, detailed textures, cinematic lighting.
**14. Princess, jellyfish crown, mysterious underwater cave, glowing crystals, ethereal light, sense of wonder, 8k resolution.**
15. Underwater princess, jellyfish crown, surrounded by schools of luminous fish, vibrant coral, crystal-clear water, cinematic lighting.
**Category: Alternative Headwear Interpretations:**
16. Close-up portrait of a princess with a headdress made of intricately woven sea anemones and jellyfish, underwater, ethereal lighting.
17. Princess, ornate headdress inspired by jellyfish, intricate details, flowing fabric, underwater, cinematic lighting, 8k resolution.
18. Portrait of a princess with a crown sculpted from fossilized jellyfish, underwater ruins background, mysterious atmosphere.
19. Princess with a headdress resembling a blooming jellyfish, vibrant colors, detailed textures, underwater, soft lighting.
20. Close-up of a princess wearing a delicate headdress crafted from translucent jellyfish, ethereal glow, underwater scene, serene expression.
**Category: Completely Different Prompts (Rephrasing):**
21. A breathtaking underwater portrait of a regal woman, adorned with a luminous, organic headpiece, evoking a sense of otherworldly beauty.
**22. An ethereal underwater scene depicting a queen, her crown formed by bioluminescent sea creatures, radiating an enchanting glow.**
23. A captivating image of a majestic figure submerged in an ocean, wearing a radiant headpiece reminiscent of jellyfish, in a hyperrealistic style.
24. A mesmerizing underwater portrait, featuring a serene woman with a crown made of glowing jellyfish, showcasing intricate details and vibrant colors.
25. An elegant woman, her head crowned with a dazzling display of bioluminescent jellyfish, creating a stunning underwater spectacle.

**User**
User selects prompt number 22:
**"An ethereal underwater scene depicting a queen, her crown formed by bioluminescent sea creatures, radiating an enchanting glow."**

