# OpenReview forum: "Preference Adaptive and Sequential Text-to-Image Generation"
_ICML.cc/2025/Conference — ICML 2025 poster_

### Official Review · Reviewer_z2d3 · 2025-03-07

**Overall Recommendation:** 4

**Summary:**

The paper proposes a novel method for text-to-image (T2I) generation that adapts to user preferences over the course of multiple turns. The problem is framed as a Markov decision process (MDP) where the initial state is some prompt $p_0$ provided by the user, each subsequent state is given by the history of interactions up to the current turn (we assume that there are at most $H$ turns), the action is to propose a selection of $L$ text prompts for the next batch of images ($M$ images are generated for each prompt), and the transition is induced by the user's choice of his/her preferred prompt among the proposed selection. The reward is given by a _user utility function_, which quantifies the user's satisfaction with a given prompt based on the images that were generated with it.

The task now becomes to maximize the cumulative reward, i.e. the user utility, over the course of $H$ turns by proposing an appropriate set of prompts in each turn. To this end, an agent is trained with offline reinforcement learning (implicit Q-learning) to select the best $L$ prompts from a set of $L_C$ candidate prompts generated using a fixed multi-modal language model (Gemini 1.5 Flash). Training data is obtained using a baseline prompt generator that interacts with human annotators who, for each turn, label their preferred prompt among the presented slate. In order to quantify the _utility function_ encoded in these preference labels, a _user model_ is trained on the gathered data as well as preference datasets from the literature. The user model's objective is to predict which prompt was chosen as the preferred one by the human annotator. To bulk up the dataset, a simulated dataset of user interaction is generated by using the trained user model as a proxy for human annotators. Finally, the prompt selection agent is trained using offline RL where the reward signal is obtained from the user model.

Evaluation consists of three components: The user model's accuracy on human preference prediction, the user model's accuracy on user choice prediction, and the selection agent's ability to adapt the selected prompts to the user's preference. Positive results are reported in each case.

### Update after rebuttal

After reading the other reviews and the authors' rebuttals, and taking into account the additional experimental results, I believe that the paper proposes a good idea and executes it well. Of course there is still room for improvement: ablating the importance of the LLM and/or prompt as well as improving the clarity of the writing comes to mind. However, if the promised clarifications can be implemented, and given the significance of the investigated problem and the potential impact of releasing the full training data, I believe that the paper meets the acceptance threshold. To reflect this, I will increase my score from 3 (weak accept) to 4 (accept).

**Claims And Evidence:**

The paper’s claims are generally well-supported, with a couple opportunities for improvement.
1. The fact that the proposed agent is trained with offline RL (as opposed to online) is worth highlighting more clearly, as this is a big distinction.
2. The paper claims that there are two separate models, one user _utility_ and one user _choice_ model (L239 ff.), but from what I can tell, there is only a single model being trained (the preference model). Besides the added temperature parameter, is the user choice model a different model?
3. The reported 99% confidence intervals in Figure 5 may be overly pessimistic. As presented, all confidence intervals overlap, making the results not statistically significant, which, if it’s the case, should be discussed. A different statistical test may offer a higher significance level.
4. Releasing the code and trained models in addition to the dataset would go a long way to bolster the paper’s claims, as it resolves any concerns about reproducibility stemming from somewhat limited details/clarity on the model architecture and training procedure.

**Essential References Not Discussed:**

The background of preference-adaptive image generation is covered fairly well. Additionally, there exist methods for iterative prompt improvement based on human feedback [1] or based on automated scoring [2] that may be worth mentioning.
- [1] Martins et al., 2023. https://cdv.dei.uc.pt/wp-content/uploads/publications-cdv/martins2023metaprompter.pdf
- [2] Mañas et al., 2024. https://arxiv.org/abs/2403.17804

**Experimental Designs Or Analyses:**

The conducted experiments seem sound besides what is already mentioned above.

**Methods And Evaluation Criteria:**

The conducted experiments make sense for evaluating both the user model as well as the T2I agent, and the use of human raters for evaluation is a big strength.
1. Since we have a model-based way to measure user preference, it would have been nice to see an improvement over multiple turns in that regard as well. E.g. what is the average satisfaction/improvement according to the user model in each round, and does it saturate or keep increasing?
2. Given that there already exist methods for multi-turn text-to-image generation (e.g. von Rütte et al., 2023; Liu et al., 2024), some sort of comparison (qualitative or quantitative) would feel appropriate. What does the proposed method bring to the table that previous approaches didn’t?

**Other Comments Or Suggestions:**

Nits:
- Line 31, Column 2, also other occurrences: Gemini report is referenced as “Team et al.”, should be “Gemini Team”.
- Line 45, Column 2: missing period after “LMM”
- Section 2.1, RL: The value function, strictly speaking, measures the expected cumulative reward of a given state, not state and action (this would be the state-action value function, or Q function).
- Footnote 4 is of little value without further details about the contractors (e.g. country/method of employment country, compensation)
- Eq. 3: Why introduce a generic aggregator if we are anyways only going to use softmax?

**Other Strengths And Weaknesses:**

Strengths:
- The paper tackles an important problem in text-to-image generation in trying to reduce the burden of prompt engineering, continuing a line of work that consists of incorporating user feedback to adapt the generation process.
- The proposed approach to do this is novel and makes sense, leveraging value-based RL to train a user-adaptive prompting agent.
- Releasing the training data will allow future work to build and improve on the results of this work.
- Evaluating the trained model using human raters directly measures the downstream performance.

Weaknesses:
- Clarity of writing is the paper’s biggest weakness. This includes multiple separate aspects:
  - Flow: It may be helpful to switch the order of sections to more accurately reflect the logical flow of the project, which is dataset creation -> user model training -> synthetic data generation -> offline RL training -> evaluation (consisting of user model evaluation -> agent evaluation). As is, the paper (to me) felt somewhat out-of-order, and it took me a couple of passes to wrap my head around it.
  - The formalism introduced in Section 2.2 feels a bit contrived and unnecessarily sophisticated. This may obfuscate the actual problem formulations and confuse some readers, a clear and simple textual introduction may be more appropriate. For an MDP, we just need to know the state space, the action space, and the reward function. For the sake of Section 2.2, it may suffice to state this in natural language.
  - Important details are deferred to the appendix (presumably due to lacking space caused by, IMO, unnecessary formalism), leading to obfuscation. The model architecture and training is a core contribution of the paper and should be covered properly in the main text. In fact, I would argue that all of the formalism introduced at the beginning of Sections 2 and 5 is fluff that should be replaced with the actual meat that is the design and training of the user model and prompting agent. This is what the paper is about, not whether the user actually does or does not choose the image that they prefer most (Section 5).
  - The lack of clarity somewhat extends to the evaluation setup: How can accuracy on Pick-a-Pic and Spearman correlation on HPS be computed if the user type is considered given in the user model? How is the user type determined at inference time? Or, when evaluating PASTA, is the preferred prompt chosen by the human rater or by the user model?
  - Minor questions regarding clarity are deferred to the “Questions For Authors” section.
- Not releasing the code and trained models is closely behind as the paper’s second biggest weakness. As-is, I would not feel confident in being able to reproduce the results presented in the paper, due to lacking details and/or vague/confusing writing (just one example: the main text uses $R$ to denote user utility, but the same quantity seems to be called $s$ in the appendix).
- The prompt generation/expansion module of the proposed agent is a significant factor that potentially has major impacts on performance and warrants dedicated evaluation/ablation, or at the very least discussion. For example, does the proposed way of sampling 5 prompts from 5 different categories actually help? And if so, by how much? Other things like the “system prompt” (which is chosen arbitrarily), model, output format, etc. can also plausibly affect the performance of the final agent in a significant way.
- Impact statement: The field of T2I modeling has a potentially big impact on the economics of visual content creation. Agentic content recommendation also has potential risks associated with it. Providing some perspective on this feels appropriate.

Given the relevance of the tackled problem and the strength of the contributions, I am inclined to recommend an accepting decision. However, considering the clarity of the writeup and the mentioned concerns regarding reproducibility, I cannot give a strong recommendation.

**Questions For Authors:**

1. Will the model weights and training code be released?
2. How is the human-rated sequential data generated? Specifically, how are the prompt slates generated and, precisely, how is the slate selected from candidate prompts? It feels like this would already require an agent, which, presumably, we don’t have yet.
3. In what sense is the user model _not_ simply a reward/preference model? This term is much more common in the literature and very easy to understand, so if appropriate, it should be preferred.
4. What is the setup and sample size for human evaluation of the final agent?
5. L302, Col. 2: “Rewards provided after the final round” – what are the rewards? Based on a model or based on human raters?
6. Assumption 5.2: What does it mean for Equation 2 to be satisfied? Equation 2 seems to be an expression, not an equality.
7. What are baseline performances on Pick-a-Pic and HPS? Without even the scores from the original papers, the presented numbers are rather meaningless.
8. Figure 3: What is the shaded area? X- and y-labels are missing. Unclear what “cross-turn accuracy” is.
9. L304, Col. 2: What does “softmax sampling” mean? Do we sample a random index based on the softmax distribution? Or do we compute a weighted average over the scores, with the weights given by the softmax distribution?
10. L306, Col. 1: How does the temperature parameterization “ensure” that Assumption 5.2 is satisfied? Did we not make this assumption beforehand in order for this parameterization to be valid?

**Relation To Broader Scientific Literature:**

The paper introduces an adaptive prompt-refinement agent that is trained using offline RL, which to the best of my knowledge has not been done before and is a valuable addition to the literature of both prompt-refinement and personalization through iterative feedback. In terms of performance, it is not clear how the proposed method stacks up to existing approaches, and by extension, whether it is worth the fairly considerable increase in complexity.

Besides the proposed model, the collection and release of iterative preference data is a valuable contribution to the field that may lay the foundation for future work.

**Theoretical Claims:**

As far as I can tell, there are no major theoretical claims.

---

> ### Author Rebuttal · Authors · 2025-04-01
>
> We appreciate the reviewer’s helpful feedback, as well as their recognition of the significance and originality of our work and the value of sharing our distinctive dataset. Below, we respond to the reviewer’s comments.
>
> **A: Model weights and training code release**
>
> We are currently reviewing options for releasing our user model in addition to the already open-sourced complete datasets.
>
> **B: “...proposed agent is trained with offline RL (as opposed to online) is worth highlighting more clearly..”**
>
> This is a good point. Offline RL was chosen because: 1) It allows (also) training directly on the rich human rater dataset, capturing authentic complexities. Online RL with real people is impractical, forcing reliance on simplified user models. 2) Offline RL is faster, avoiding slow environment interactions and inference with large user models/LLMs required by online RL. We will highlight this.
>
> **C: “...is the user choice model a different model?”, “the main text uses $R$ to denote user utility, but the same quantity seems to be called $s$ in the appendix”**
>
> In the paper we assume equivalence assumption between choice and preference and consistency in utility. A trained score function $s_\theta$ builds the utility function (Eq 3), scoring image slates (per prompt/user type). Applying softmax over utility scores (Eq 4) yields a user choice probability consistent with utility and the preference model.  A learned temperature adds flexibility while keeping the assumptions (scaling the entire vector of utility scores by a constant $\tau_\theta$ does not alter the ranking consistency). Thus, both models share $s_\theta$ with the choice model derived from utility output. $R$ refers to the overall utility concept, while $s$ is the specific learned score function implementing it.
>
> **D: “The reported 99% confidence intervals in Figure 5 may be overly pessimistic”**
>
> Confidence intervals don't overlap vs. the base model, but some overlap exists when comparing our method with different data types. We'll run another rater study assessing full trajectory improvement (final vs. first turn), expecting clearer results as cumulative improvement is what we ultimately care about. Updates during discussion.
>
> **E: “How can accuracy on Pick-a-Pic and Spearman correlation on HPS be computed if the user type is considered”**
>
> The test sets for Pick-a-Pic and HPS are designed such that each sample contains a specific annotator's samples. To assess the posterior of the annotator over user types, we sampled a subset (of size 3) from the full set of annotated samples, then calculated the accuracy or rank of the remaining held-out samples and averaged the results over the posterior. We found this comparable to using the full set for the posterior computation. This detail was omitted due to space and it will be added in revision.
>
> **F. Questions For Authors:**
> 1. See comment A.
> 2. For human data creation, we used Gemini 1.5 Flash as the candidate generator. Instead of a trained selector, a random selector proposed the slate, picking up to one prompt per category from the generator. This combination provides quality expansions and broad exploration, crucial for building an unambiguous offline latent MDP dataset.
> 3. Our user model consists of the utility component as well as the user choice model that translates this reward into user choices, as is standard in utility theory, choice modeling, econometrics, etc. Indeed, for a fixed choice model the user model is a generalized (user specific, set-based) preference model.
> 4. Our evaluation consists of 60 human raters, each experiment was evaluated over 200 trajectories. Exact metrics will be in the appendix.
> 5. Agent trained with sparse rewards (only at final 5th round), determined by the learned user utility function. Human rater data was also labeled using this same learned utility function post-collection.
> 6. Eq 2 is the generalized preference probability of preferring image set $\ell^*$ over the other sets. This is not an equality but a description of the preference probability.
> 7. Since the user model's goal is mimicking users for data generation, our metrics focus on performance across user types (averaged over posterior). However, we will add original baseline scores from the Pick-a-Pic/HPS papers for comparison.
> 8. The y-axis represents the accuracy outlined in the subtitles, while the x-axis indicates the number of user types used in the user model. The shaded area is 95% CI. We will clarify this in the paper. Appendix D.3 details cross-turn accuracy: preference accuracy between selected images at consecutive test set turns.
> 9. Softmax sampling: The Eq 3 aggregation function can be max, average, softmax sampling, etc. Softmax sampling means selecting an element randomly based on the softmax distribution over scores. We chose it over max to add randomness reflecting user selection variability observed during data collection and sequential T2I.
> 10. see comment C.

---

### Official Review · Reviewer_LRxo · 2025-03-12

**Overall Recommendation:** 2

**Summary:**

This paper introduces PASTA, a reinforcement learning (RL) agent designed for preference-aligned, sequential text-to-image (T2I) prompt expansion. The framework aims to enhance T2I generation by formulating multi-turn interactions as a sequential decision-making problem, leveraging LMMs (Gemma) and image diffusion models (Stable Diffusion XL). The paper evaluates PASTA through human and simulated experiments, claiming improved user satisfaction and model-agnostic adaptability.
Main Contributions:
1. Formulate the challenge of multi-turn T2I generation as a sequential decision-making problem.

2. The use of IQL for offline RL and user category-based prompt generation is innovative, enhancing diversity and personalization.

3. Use LMM(Gemma) and the Image Diffusion Model (SD-XL) to construct the generation framework.

4. Collect sequential sample rater data and simulated data for multi-turn T2I generation and facilitating further research.

**Claims And Evidence:**

1.In section 7, “PASTA operates by directly adapting the input prompt, offering a model-agnostic approach to adaptive preference learning in T2I generation”. Could you provide more evidence for this? As different T2I models may perform different given the same prompt settings, also LMM’s capability could affect this prompt expansion process. The lack of experiments with other SOTA T2I generators (e.g., DALL-E 3) or LMMs weakens this claim. Additional validation across diverse models is necessary to substantiate model-agnostic adaptability.

2.Presented User Model improves user satisfaction compared to baseline(Gemini 1.5 Flash). Could you compare your with other preference-alignment reward models (e.g. ImageReward), T2I refiner(G-Refine, Idea2Img) or more advanced LMMs (e.g. GPT-4o)?

**Essential References Not Discussed:**

None

**Experimental Designs Or Analyses:**

1. Human evaluations rely on subjective turn-over-turn judgments (Better/Worse/Same), lacking quantitative and objective metrics like image quality scores (e.g., FID, IS). This limits the assessment of PASTA’s performance in terms of visual fidelity and semantic accuracy, which are critical for T2I tasks. We may need more experiments in image generation benchmarks evaluating both preference and objective metrics like DreamBench++ to show that your framework could generate images having significantly better performance than baseline in more objective metrics.

2.Better / Worse / Same selection results between different turns could be influenced a lot by psychological factors, only by sampling different images, users may also select “Better” at a probability, even if there is no significant prompt changes between different turns. The paper does not provide “blank” experiments results to measure this bias.

**Methods And Evaluation Criteria:**

1. Strict word limits (12 for human, 10 for simulated) may not reflect real-world usage, limiting applicability.

2 . What is the intention of not allowing users to see intermediate prompts?

**Other Comments Or Suggestions:**

I am willing to raise my score if the authors can address my concerns.

**Other Strengths And Weaknesses:**

None.

**Questions For Authors:**

Illustrations (e.g., "An image of happiness/love," Figure 6) focus on abstract prompts, which may not be representative enough of typical T2I applications. How do these prompts align with real-world usage, and what is PASTA’s performance on different categories of prompts?

**Relation To Broader Scientific Literature:**

N/A

**Theoretical Claims:**

1. What is the essence of recognizing user’s category or type in a multi-turn generation process instead of explicitly using textual interactions? For example, if a user wants to have realistic and scenic images about a Railroad Journey (Example #1 in Appendix G), why couldn’t we lay out several generation styles for choosing and then generate more style-aligned images efficiently?

---

> ### Author Rebuttal · Authors · 2025-04-01
>
> We thank the reviewer for their detailed review and hope that the following response addresses the reviewer’s concerns.
>
> **A: "The lack of experiments with other SOTA T2I generators.."**
>
> **Response:** To address this, we are running additional test-time experiments evaluating models our agent wasn't trained on. This includes a rater study using our method with Flux.1 and comparing our method to a GPT4.5-based agent. Results will be shared during the discussion period.
>
> **B: "Could you compare your with other preference-alignment reward models?", "lacking quantitative and objective metrics"**
>
> **Response:** We agree further analysis is helpful.  We will add additional evaluation metrics to our experiments during the author-reviewer discussion phase (by next week) including: user model score, FID, and LPIPS.
>
> **C: “Strict word limits..”**
>
> **Response:** The constraint arises from the pretrained CLIP text encoders used in the user-score model, which are limited to 77 tokens. To stay below this threshold after five turns, we set the initial prompt to 12 words. Additionally, the text-to-image (T2I) model has a limited context capacity, restricting the level of detail that can be included. Despite this, the model still accommodates a variety of starting prompts that can evolve into detailed and rich expansions. Moreover, keeping initial prompts short shifts the focus toward feedback derived from user choices rather than critiques, which aligns more naturally with generalizing the standard single-turn preference setup. We intentionally minimize the influence of critiques to maintain this balance. That said, using improved encoders like LLM2CLIP is left for future exploration.
>
> **D: “What is the intention of not allowing users to see intermediate prompts?”**
>
> **Response:** PASTA refines prompts preserving user intent. Users choose the prompt-image pair best matching their initial prompt, preserving core elements. This method keeps the sequence of generated images closely tied to the content of the starting prompt, reducing the likelihood of producing images that stray too far from the user's starting point (e.g., beginning with a prompt about a train and ending up with an image of a dog). We experimented with displaying intermediate prompts to raters but found that presenting these prompts especially given their often complex nature placed too great a cognitive load on raters. We also believe this could compromise the quality of feedback provided by raters. Prompt refinement and expansion are natural (and commonly used) interaction modes for users engaged in iterative image generation. Surfacing the prompts as suggestions along with the images is something to be explored, but might require more "controlled" prompt expansion to elicit proper responses. We will elaborate further in our paper.
>
> **E: “What is the essence of recognizing user’s category or type in a multi-turn generation process instead of explicitly using textual interactions?”**
>
> **Response:**  PASTA doesn't identify user types explicitly. Types only simulate consistent preferences in synthetic data creation. The agent adjusts to the user by leveraging the interaction history, adapting to their preferences. Our method offers advantages over textual instructions because users are often unaware of the prompt possibilities provided by the agent, and selecting images is significantly more user-friendly than drafting instructions.
> That said, PASTA can incorporate textual instructions by integrating them into the candidate generator’s prompt, allowing it to focus on specific categories or directions specified by the user. Our dataset includes rollouts (20%) with text critiques for potential model training. We kept the framework simple, leaving text integration for future work.
>
> **F: “Better / Worse / Same selection results between different turns could be influenced a lot by psychological factors”**
>
> **Response:** User feedback inconsistency due to psychological factors is expected. However, average human evaluations remain a valuable performance measure. Our results show the trained PASTA agent significantly outperforms a standard LLM.
>
> **G: “Is the “sample rater data” provided in the supplementary material the full version of your curated human rater data?”**
>
> **Response:** Our human rater data, collected with a random policy selector, consists of over 7k 5-step rollouts (>500k images), hundreds of prompts, approx. 100 raters. Simulated data: over 30k rollouts (>2.5M images). The supplementary sample is small due to size limits. A link to the full dataset will be provided in the final version of the paper.
>
> **H: “How do these prompts align with real-world usage..”**
>
> **Response:** Abstract prompts help discern user type preferences during interactions better than typical prompts (e.g., 'dog running'), which often have universally preferred styles and biases. Supplementary materials include full rollouts for typical T2I prompts.

---

### Official Review · Reviewer_udgs · 2025-03-12

**Overall Recommendation:** 3

**Summary:**

This paper introduces PASTA, a RL framework for interactive text-to-image (T2I) generation.
This method enables multi-turn collaboration between models and users to refine prompts/images iteratively to align the preferences of users.
A novel dataset of sequential user interactions is proposed and a user simulator trained via EM strategies to model diverse preference types. Evaluations show PASTA outperforms baseline models in human ratings
This paper compared the situations of training on combined real and simulated data or training on sole dataset.

## update after rebuttal
The provided additional evaluation addressed most of the reviewer's concerns thus the reviewer choose to enhance the rating to 3.

**Claims And Evidence:**

Refer to the questions in "Experimental Designs Or Analyses".

**Essential References Not Discussed:**

No.

**Experimental Designs Or Analyses:**

The article only conducted experiments on Gemini + SDXL, failing to demonstrate the generalizability of the method.

Why was the number of interaction turns set to five? According to Figure 5, the results continue to improve even in the fifth turn, necessitating a convergence analysis.

This paper should evaluate more metrics, in addition to Pick-a-Pic and HPS accuracy, other metrics such as FID (for assessing quality) and LPIPS (for measuring diversity) could also be evaluated.

**Methods And Evaluation Criteria:**

Strength:
Prompt rewriting is essential for text-to-image generation. This paper introduces the first RL-based multi-turn interactive text-to-image generation framework PASTA, mainly for generating a better prompt for T2I that can progressively refines the generated results through dynamic prompt expansion.

This paper constructed the first dataset that encompasses multi-turn interactive behaviors of human evaluators.

Weakness:
This method drawback lies in the costly data collection process. This paper does not mention the scale of user annotations. If a different generative model is used, would the same annotations be required? The reviewers are concerned that the annotation scale might be too large, costly and not reusable.

This paper does not analyze whether the synthetic data contains noise, the extent of such noise, or the overall quality of the data. Whether there exists a domain gap between single-turn and multi-turn preferences for models trained on single-turn preference datasets requires thorough analysis.

Generalizability of the method. It remains to be verified whether this approach would still be effective if applied to other LMMs (Large Multimodal Models) and visual generators. Additionally, the generalizability of the trained value model (in this case, Gemma) needs to be assessed.

**Other Comments Or Suggestions:**

Conduct experiments on a wider array of generative models.

**Other Strengths And Weaknesses:**

No.

**Questions For Authors:**

Could the authors perform a manual or automated analysis of the synthetic data's quality?

**Relation To Broader Scientific Literature:**

This paper holds significant implications for the application of reinforcement learning in the realm of text-to-image synthesis.

**Theoretical Claims:**

The theoretical claims are sound.

---

> ### Author Rebuttal · Authors · 2025-04-01
>
> We are grateful for the reviewer’s constructive feedback and for pointing out our dataset contribution as the first T2I dataset for multi-turn interaction setting. Bellow we address the concerns raised by the reviewer:
>
> **Comment 1: "This paper does not mention the scale of user annotations."**
>
> **Response:** Thank you for highlighting this oversight. Our human rater data consists of over 7,000 five-step rollouts (totaling more than 500,000 images), annotated by approximately 100 human raters using a random candidate selector. The simulated user data includes over 30,000 rollouts (exceeding 2.5 million images). We will add this information to the paper.
>
> **Comment 2: "The reviewers are concerned that the annotation scale might be too large, costly and not reusable."**
>
> **Response:** To encourage research in this novel setting, we are making all datasets used during training publicly available—we hope this will allow researchers who lack the resources to conduct such studies to benefit. Furthermore, we have proposed that exploiting trained user simulators, independent of large language models and easily trainable with the provided dataset, is an effective solution to mitigate data scarcity. Finally, we are sharing the synthetic dataset created by our user simulator to further assist researchers in addressing these challenges with greater ease.
>
> **Comment 3: “Generalizability of the method.”**
>
> **Response:** Thank you for raising this point. To address this we are now running additional experiments with different models during test-time (i.e., evaluating using models that our agent was not trained on). Specifically, we are conducting an additional rater study using our method with the Flux.1 T2I model. We are also running a study to compare our method to a GPT4.5-based model as an agent. We will comment here with these new results during the discussion period.
>
> **Comment 4: Analyze the synthetic data.**
>
> **Response:** Our approach to analyzing the synthetic data involves assessing the user model, which represents the sole distinction between the real human data and the data produced by the simulated user model. Visually, we can observe a clear distinction between user types by either scoring the top five highest-rated images in the HPS test set or by reviewing the rollouts of PASTA against the user model, both of which highlight distinct preferences across different user types. For a numerical perspective, we refer to Section 6.2, particularly Figure 3, where our user model achieves a prediction accuracy of 70% compared to real human raters, suggesting that the synthetic dataset effectively captures essential aspects of real user interactions. However, we agree with the reviewer’s perspective that further analysis is necessary. As such, we will add additional evaluation metrics to our experiments, including: user model score, FID, and LPIPS. We will comment here with these new results during the discussion period.
>
> **Comment 5: “...domain gap between single-turn and multi-turn preferences..”.**
>
> **Response:** Indeed, there is a distributional shift between single-turn and multi-turn preferences. However, our user model parameterization, encompassing both the user choice model and utility, relies on a single-turn score function $s_\theta$. Consequently, we can utilize single-turn datasets like Pick-a-Pic to augment the data available for training the user model. That said, due to the distributional disparity and our emphasis on the multi-turn context, these single-turn datasets are employed solely in the initial phase of the user model training to provide a stronger starting point for the subsequent phase training with the human rater multi-turn dataset, which is our primary focus. This point may not be entirely transparent in our exposition: we will elaborate further in our discussion of the user model training process in the paper.
>
> **Comment 6: “Why was the number of interaction turns set to five?”**
>
> **Response:** We settled on five steps to balance between providing a challenging enough interaction length and meeting the real-time expectation of users for relatively swift engagement with the agent. Exploring the (variable) optimal rollout lengths is something we have deferred to future research, given that the PASTA framework includes numerous components that warrant deeper study.

---

### Official Review · Reviewer_ym4T · 2025-03-17

**Overall Recommendation:** 3

**Summary:**

This paper introduces PASTA, a novel reinforcement learning framework for interactive text-to-image generation. It addresses the challenge of capturing precise user intent through iterative prompt expansion. The core ideas involve using a large multimodal language model (LMM) for prompt candidate generation, a value-based RL agent for prompt selection, and a user model trained on both human and simulated data to guide the agent. The paper also contributes a new dataset of sequential user preferences.

**Claims And Evidence:**

The claims are generally well-supported by experimental evidence. The human evaluation demonstrates a significant improvement in user satisfaction compared to baseline methods, particularly when PASTA is trained on a combination of real and synthetic data. However, the performance gains from the user simulation require further scrutiny.

**Essential References Not Discussed:**

NAN

**Experimental Designs Or Analyses:**

The experimental design appears sound, with appropriate baselines and evaluation metrics. The analysis of the impact of different training data regimes is insightful. More discussion is needed on limitations on rater bias and diversity.

**Methods And Evaluation Criteria:**

The proposed methods are well-suited for the problem of interactive text-to-image generation. The decomposition of the problem into candidate prompt generation (using a large multimodal language model) and candidate selection (using value-based reinforcement learning) is a sensible approach. The LMM allows for exploration of a diverse set of prompt expansions, while the RL agent learns to choose the most promising prompts based on user feedback. The implicit Q-learning (IQL) algorithm is a reasonable choice for offline RL in this context, given its ability to handle overestimation issues.

The evaluation criteria are also appropriate. The human evaluation, where raters judge the improvement of images across turns, directly measures the effectiveness of the interactive refinement process.  The use of both absolute satisfaction scores and turn-over-turn comparisons provides a comprehensive assessment of user experience.  The simulated user experiments offer a valuable means to analyze the impact of different training data regimes and reward settings.

However, the reliance on specific, potentially proprietary, pre-trained models might raise concerns about reproducibility and accessibility.  It would be beneficial to discuss the potential impact of model choice on the overall performance.

**Other Comments Or Suggestions:**

Consider adding a section discussing the limitations of the user simulation and potential biases it may introduce.

**Other Strengths And Weaknesses:**

Strengths: The paper tackles an important problem with a novel and well-designed framework. The dataset contribution is valuable. The use of both human and simulated data is clever.
Weaknesses: The gains from the user simulation require further justification. The dependency on particular pre-trained models(Gemini, Gemma, SDXL) might limit the generality of the results.

**Questions For Authors:**

Could you provide a more detailed analysis of the specific dynamics captured by the user simulator that contribute to the performance gains observed in the simulated user experiments? (Understanding this better would strengthen the justification for using simulated data).
What are the computational costs for implementing and training PASTA?

**Relation To Broader Scientific Literature:**

This paper builds upon recent advances in interactive and preference-adaptive image generation, particularly those leveraging LLMs and RL. It extends existing methods by framing multi-turn image generation as a sequential decision-making problem, allowing for iterative refinement towards a desired visual outcome.

**Theoretical Claims:**

No Obvious Problem.

---

> ### Author Rebuttal · Authors · 2025-04-01
>
> We appreciate your positive feedback on the novelty of our approach and our dataset contribution for the community. Bellow we address the concerns raised by the reviewer:
>
> **Comment 1: "...the performance gains from the user simulation require further scrutiny."**
>
> **Response:** The user simulator is used for two reasons: (1) generating synthetic data that is user-coherent in order to extend the real human rater dataset; and (2) labeling the human rater dataset. Our experiment with real users shows that adding additional synthetic data improves the model’s performance, indicating that our generated dataset is beneficial for training in real-world scenarios. In addition, we examine the ability of our user model to predict real users choices with 70% accuracy (Fig 3), which further indicates that our user simulator does capture some key dynamics of real user interactions. We will add a plot showing the user model score over time in order to add an additional performance evaluation metric. We will comment here with these new results during the discussion period.
>
> **Comment 2: "The dependency on particular pre-trained models(Gemini, Gemma, SDXL) might limit the generality of the results."**
>
> **Response:** Thank you for raising this point. In order to address it we are running additional experiments with different models during test-time (i.e., evaluating using models that our agent was not trained on). Specifically, we are running an additional rater study using our method with the Flux.1 T2I model. We will also compare our method to the use of the GPT4.5 model as an agent. We will comment here with these new results during the discussion period.

---

### Decision · Program_Chairs · 2025-05-01

**Decision:**

Accept (poster)

**Comment:**

This paper proposes a novel reinforcement learning framework named PASTA for interactive text-to-image generation. The final recommendations are 1 "Weak reject", 2 "Weak accept",  and 1 "Accept". The reviewers generally agree that
- the problem is important,
- the formulation is novel,
- claims are well-supported by experiments,
- evaluation criteria are appropriate, and
- dataset contribution is valuable.

Their major concerns include
- clarity of writing, and
- generalizability.

The authors povided additional results after the rebuttal to address the concern on generalizability.

I recommend "Accept" based on the importance of the problem and the novelty of the formulation.